# Neurophysiological sensitivity to envelope and pulse timing interaural time differences in cochlear implanted rats with different hearing experiences

S. Fang[1,2] (iD), T. Fleiner[3,4] (iD), F. Peng[2], S. Buchholz[3,4] (iD), M. Zeeshan[1,2], N. Rosskothen-Kuhl[3,4,5] (iD) and J. W. Schnupp[1,2,6] (iD)

[1]*Department of Neuroscience, City University of Hong Kong, Hong Kong SAR, China*
[2]*Gerald Choa Neuroscience Institute, Chinese University of Hong Kong, Sha Tin, Hong Kong SAR, China*
[3]*Faculty of Biology, University of Freiburg, Freiburg, Germany*
[4]*Neurobiological Research Laboratory, Section for Experimental and Clinical Otology, Department of Otolaryngology, Medical Center – University of Freiburg, Medical Faculty, University of Freiburg, Freiburg, Germany*
[5]*Bernstein Center Freiburg and Faculty of Biology, University of Freiburg, Freiburg, Germany*
[6]*Department of Otolaryngology, Chinese University of Hong Kong, Hong Kong, China*

Handling Editors: Nathan Schoppa & Conny Kopp-Scheinpflug

The peer review history is available in the Supporting Information section of this article (https://doi.org/10.1113/JP290143#support-information-section).

**Abstract figure legend** Graphical summary of our study. Rats with varying hearing experiences were bilaterally implanted with cochlear implants (CIs), and neural responses were recorded from the inferior colliculus (IC), an auditory

S. Fang and T. Fleiner shared first authorship.

This article was first published as a preprint. Fang S, Fleiner T, Peng F, Buchholz S, Zeeshan M, Rosskothen-Kuhl N, Schnupp JW. 2025. Neurophysiological Sensitivity to Envelope and Pulse Timing Interaural Time Differences in Cochlear Implanted Rats with Different Hearing Experiences. bioRxiv. https://doi.org/10.1101/2025.09.11.675523

The Journal of Physiology

midbrain region (left). IC neurons showed greater sensitivity to interaural time differences (ITDs) carried in pulse timing (thumbs up; right top) than in the envelope (thumbs down; right bottom), regardless of pulse rate, modulation rate or hearing experience.

**Abstract** Cochlear implants (CIs) have successfully restored hearing in more than one million patients with severe to profound hearing loss worldwide. Although CIs effectively restore speech perception in quiet environments, sound localization remains challenging for bilateral CI users, particularly their ability to utilize interaural time differences (ITDs). The majority of clinical CI processors use a coding strategy that encodes ITD information only in the envelope of electrical pulse trains rather than their pulse timing, which may contribute to the poorer spatial hearing perception of CI users. We recently demonstrated in a behavioural study on early deafened, bilaterally CI-implanted rats that pulse timing ITDs completely dominate ITD perception, whereas sensitivity to envelope ITDs is almost negligible in comparison. Building on this, we here investigated the neurophysiological sensitivity of the inferior colliculus (IC) to envelope and pulse timing ITDs at two different pulse rates (900 and 4500 pulses/s) and three different stimulation modulations (5, 20 and 100 Hz) in CI rats with different hearing experiences. Our results indicate that IC neurons exhibit far greater sensitivity to pulse timing ITD than envelope ITD independent of pulse rate, modulation rate or hearing experience. These findings suggest that to improve binaural hearing outcomes in bilateral CI users, clinical stimulation strategies should provide informative pulse timing ITDs.

(Received 19 September 2025; accepted after revision 27 March 2026; first published online 22 April 2026)
**Corresponding authors** N. Rosskothen-Kuhl: Faculty of Biology, University of Freiburg, Freiburg, Germany. Email: nicole.rosskothen-kuhl@uniklinik-freiburg.de
J. W. Schnupp: Department of Neuroscience, City University of Hong Kong, Hong Kong (SAR China). Email: jschnupp@cuhk.edu.hk

## Key points

- Current bilateral cochlear implants mostly provide time cues of arriving sound in the envelope of stimulation rather than in the timing of pulses.
- A behavioural study on cochlear implanted rats has shown that pulse timing dominates binaural hearing.
- Here we examined the neural sensitivity to independent changes in time cues presented on the envelope or pulse timing, respectively, in rats with bilateral cochlear implants and different hearing experiences.
- Auditory midbrain neurons were consistently more sensitive to changes in pulse timing rather than envelope, irrespective of stimulus pulse rate, modulation rate or an animal's hearing experience.

**Shiyi Fang** is a PhD student at City University of Hong Kong. With a background in electronic engineering she discovered her passion for neuroscience during her master's studies at City University of Hong Kong. Her research in auditory neuroscience focuses on electrical hearing through cochlear implants (CI), with emphasis on sound localization mechanisms. She specializes in electrophysiological recordings and analysis in CI-implanted rat models, aiming to improve spatial hearing in clinical CI applications. **Tim Fleiner** studied biology (B.Sc.) and subsequently neuroscience (M.Sc.) at the Albert-Ludwigs-University in Freiburg. Since September 2022 he has been a PhD student at the University Medical Centre Freiburg in the Department of Experimental Clinical Otology in the Neurobiological Research Laboratory under the supervision of Dr. Nicole Rosskothen-Kuhl. In his work he is investigating how directional hearing can be improved in bilateral cochlear implant recipients using novel stimulation strategies, conducting behavioural and electrophysiological studies in cochlear-implanted rats.

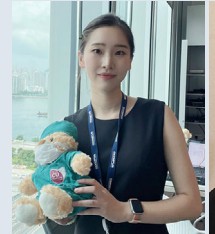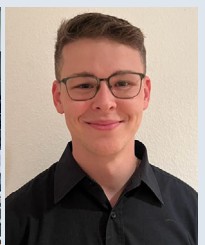

## Introduction

The advent of cochlear implants (CIs) has greatly improved the life quality of patients with severe-to-profound hearing loss by partially restoring auditory sensations, allowing these patients to understand speech and hold conservations. However the auditory performance of CI patients lags far behind that of normal hearing (NH) humans, especially in hearing tasks requiring high temporal precision, such as the use of interaural time differences (ITDs) for spatial hearing (Kan et al., 2015; Laback et al., 2004; Majdak et al., 2006; van Hoesel & Tyler, 2003). In acoustic hearing ITD information is conveyed through both the fine structure (FS) and the envelope (ENV) of the acoustic signal. The normally hearing auditory pathway can use sound arrival time differences between the left and right ear as small as 10–20 µs to discriminate sound source directions (Klumpp & Eady, 1956; Laback et al., 2007; Thavam & Dietz, 2019; Zwislocki & Feldman, 1956). In contrast ITD thresholds of bilaterally implanted CI (biCI) patients are typically larger than those of normal hearing humans, and many prelingually deaf CI patients appear to have no measurable ITD sensitivity at all (Cleary et al., 2022; Ehlers et al., 2017; Laback et al., 2015; Litovsky et al., 2010). To be precise Laback et al. (2015) found a median ITD threshold of around 144 µs testing biCI patients with specialized experimental processors at low pulse rates of around 100 pulses per second, and ITD thresholds were substantially higher at higher pulse rates (pps; Laback et al., 2015). One possible reason for the poorer ITD sensitivity of current biCI patients could be a failure of experience-dependent development of the bilateral auditory pathway due to long-term hearing impairment or the use of unilateral CI, as well as absent early hearing experience (Laback et al., 2007, 2015; Litovsky et al., 2012; Thakkar et al., 2020). However this is most likely not the main factor for the poorer ITD sensitivity, as adult-deafened CI patients also exhibit increased ITD thresholds (Cleary et al., 2022; Laback et al., 2007, 2015; Litovsky et al., 2012; Majdak et al., 2006; Thakkar et al., 2020). In addition recent studies in biCI rats have shown that poor ITD sensitivity can be avoided in a neonatally deafened (ND) auditory system when ITDs are provided with temporally precise electric pulses directly from the beginning of stimulation (Buck et al., 2023; Rosskothen-Kuhl et al., 2021). Therefore another possible cause contributing to the poor ITD sensitivity is the design of contemporary CI processors with their missing ability to provide accurate and precise ITD information via pulse timing (PT) ITD (Kan et al., 2018; Van Hoesel et al., 2002; Wilson et al., 1991). Thus little to none of the temporal FS of incoming sounds is encoded in the PT of the electrical stimuli. Instead devices rely largely, or entirely, on encoding the ENV of the sound through pulse amplitude modulation.

Consequently when biCI users receive ITD information, it is typically encoded in the ENVs used to modulate the pulse trains, rather than in the timing of the pulse trains sent to each ear. In fact commonly used coding strategies such as continuous interleaved sampling (CIS) or CIS-derived strategies tend to use pulses fired at a fixed rate independently in each ear, which results in PT ITDs that are unrelated to the acoustic stimulus ITD and may introduce significant confusion. A notable exception to this is the FS4 strategy from MED-EL, which does encode at least some temporal FS information in the pulse timing delivered to low-frequency channels.

CIS applies a multiple-frequency band-pass filter on the incoming complex sound signal and extracts the ENV information of each frequency band (Wilson et al., 1991). A set of fixed rate biphasic pulse trains modulated by extracted ENVs are then sent to the correlated electrode, respectively. Usually the CI manufacturer shifts the carrier phase of each channel a little to ensure that the adjacent electrodes will not be simultaneously activated, which may help prevent undesirable interactions between adjacent channels. The fixed pulse rate and phase of each channel make it impossible for contemporary CIs operating with CIS-based strategies to encode ITD information in the carrier pulse train of the electrical stimulus. Instead the PT ITDs delivered by these devices assume constant values that bear no relation to the sound-source ITD, as either the left or the right ear may lead by up to half an interpulse interval. It is simply a matter of chance whether, in any given moment, these random PT ITDs agree or disagree with any veridical ENV ITDs that the devices may have been able to encode. Theoretically it might be possible for the auditory system to 'extract' temporal attributes of the signal ENVs at each ear at a higher resolution than the electric pulse sample rate, and to use these to derive ENV ITD estimates independently from the PT ITDs. Whether the auditory pathway of CI patients is capable of doing so when provided with stimulation patterns typical for current clinical practice is doubtful.

Note that in the normally developed auditory pathway cochlear hair cells are known to perform a form of envelope extraction for high-frequency stimuli (Palmer & Russell, 1986), but this does not apply to patients with CIs, who generally lack normally functioning hair cells, and in whom the auditory nerve is stimulated directly. Consequently the extensive psychoacoustic literature on ENV ITD sensitivity in normal hearing can be an unreliable guide to what to expect in the CI-stimulated case (Bernstein & Trahiotis, 1992, 2002; Monaghan et al., 2015). Furthermore it is well documented that the auditory nerve has only a very limited dynamic range and resolution for resolving the amplitude of CI stimulus pulses, which is bound to limit the accuracy with which the CI-stimulated auditory pathway can extract pulse train ENVs (Zeng et al., 1998). In a

recent study (Schnupp et al., 2025) we showed that PT ITDs, and not ENV ITDs, dominate lateralization judgements of ND biCI rats when these were trained to lateralize amplitude-modulated pulse trains, with pulse rates commonly used in clinical practice (900 pps) or faster (4500 pps), and with ENV shapes that approximate the dynamics of speech sounds. This adds to the suspicion that CI pulse train ENV ITDs may not be represented in the activity of neurons in the auditory pathway of CI users with sufficient accuracy to be a very useful cue for binaural hearing.

In this study we expand on this behavioural finding and investigate how the ENV and PT ITDs of these stimuli are represented by neural discharges in the inferior colliculus (IC), and whether PT ITDs already dominate at this important midbrain auditory relay station. The IC is linked to sound localization and has been studied neurophysiologically and anatomically for more than five decades (Erulkar, 1959; Oliver & Morest, 1984). The IC is known to integrate and process bilateral auditory information, and many neurons in the IC are sensitive

to ITDs (Buck et al., 2021; Hancock et al., 2010; Skottun et al., 2001; Tillein et al., 2016). Work of our group investigated the sensitivity of IC neurons in biCI rats to small changes in ITD within the animal's physiological range, and revealed innate ITD sensitivity in the IC even in the absence of early hearing experience (Buck et al., 2021). However no previous studies have looked at fine-grained, relative ENV and PT ITD sensitivity in IC, specifically over the physiological range, in cohorts of animals with a range of different levels of binaural experience. We therefore conducted a series of neurophysiological extracellular recordings from the IC using three different cohorts with different hearing experience: (1) AD (adult deafened) rats supplied with acute biCIs, tested between P90 and P150 (Fig. 1*A*), (2) ND rats supplied with acute biCIs, implanted and tested around P90-150 (Fig. 1*B*) and (3) ND rats, which were implanted with biCIs in young adulthood, and which had been trained in a stimulus lateralization task with independently varying PT ITDs and ENV ITDs (Fig. 1*C*). Analogue multiunit activity (AMUA; Kayser et al., 2007; Schnupp et al., 2015) recorded from these

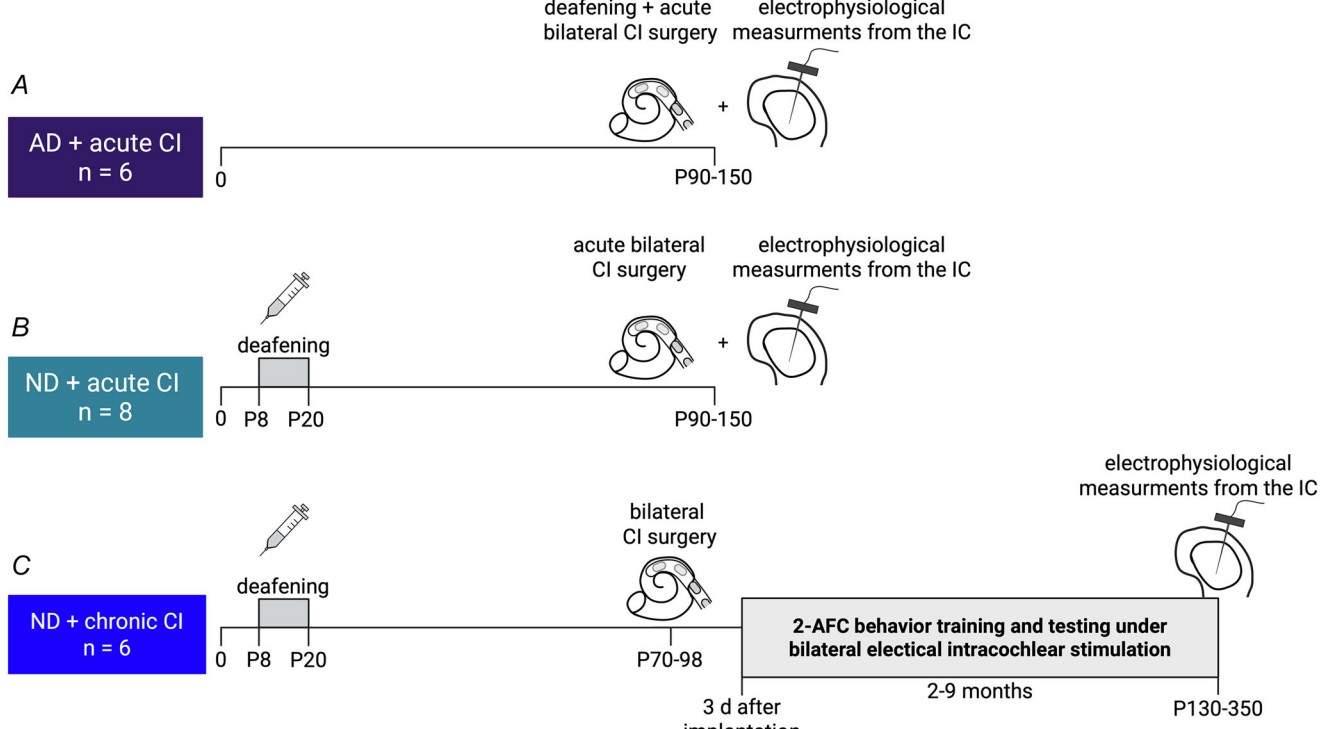

**Figure 1. Experimental timetable for the three different cohorts**
*A*, the first cohort consisted of six animals that were adult deafened (AD) and acutely bilaterally implanted with cochlear implants (CIs) immediately prior to neurophysiological recordings from the inferior colliculus (IC). *B*, the second cohort consisted of eight neonatally deafened (ND) rats. These rats were acutely supplied with bilateral CIs immediately before electrophysiological recordings were performed from IC. *C*, the third cohort included six ND animals, which were bilaterally supplied with CIs in young adulthood. From 3 days after implantation these animals underwent 2–9 months of behavioural training in two alternative forced choice tasks (2-AFC), as described in Rosskothen-Kuhl et al. (2021). After completion of the behavioural testing the rats underwent electrophysiological measurements from IC. Illustration created using BioRender.com.

animals was analysed to compute the proportion of neural response variance explained (VE) by stimulus PT or ENV ITDs, respectively, as a measure of strength of ITD tuning. The purpose of the study was to compare the distribution of VE PT and VE ENV observed at clinically relevant pulse rates (900 pps or faster) for a range of clinically relevant ENV shapes (ENV durations from 10 to 200 ms) in a number of cohorts differing in hearing experience with or without biCIs. Our prediction was that relatively strong tuning to PT ITDs would be common, and strong tuning to ENV ITDs would be rare, irrespective of stimulus conditions and cohort.

## Material and methods

### Ethical approval

All procedures involving experimental animals reported here were approved by the City University of Hong Kong Animal Research Ethics Sub-Committee and conducted under license by the Department of Health of Hong Kong (#16-52 DH/HA and P/8/2/5) and/or the Regierungspräsidium Freiburg (#35-9185.81/G-17/124, #35-9185.81/G-22/067), as well as by the appropriate local ethical review committee. We confirm that all of our methods were performed in accordance with the relevant guidelines and regulations and that our study is reported in accordance with the ARRIVE guidelines. The authors confirm that they fulfil the ethical principles under which the journal operates and that the study complies with the journal's animal ethics checklist. In total 20 female Wistar rats (150–300 g) were obtained (Janvier Labs and Chinese University of Hong Kong Laboratory Animal Service Centre) for this study and underwent terminal electrophysiological recordings. Six animals (AD + acute CI cohort; Fig. 1) were raised with normal hearing, as well as constant access to water and food until the day of acute bilateral implantation (P90–150) consecutively followed by terminal electro-physiological recording from IC. The remaining 14 rats were neonatally deafened using kanamycin injection protocols (see next section). Of these ND rats eight remained without electrical hearing experience and with constant access to water and food until the acute bilateral CI surgery and electrophysiological recording (P90–150; ND + acute CI), whereas six were bilaterally implanted in young adulthood (P70–98). Until then they had constant access to water and food *ad libitum* and received electrical stimulation while being water deprived (for details see '2-AFC training and testing of the chronic CI cohort') for 2–9 months (ND + chronic CI; Fig. 1). During this time the animals were kept in an animal cabinet with constant monitoring of the air exchange rate, temperature and humidity in a 12 h day–night cycle.

### Deafening and recording of auditory brainstem responses

ND rats were deafened by daily I.P. injections of 400 mg/kg kanamycin from postnatal days 8–20 inclusively, as described in Rosskothen-Kuhl & Illing (2012) and Rosskothen-Kuhl et al. (2018, 2024; Fig. 1). This procedure is known to cause widespread death of inner and outer hair cells (Argence et al., 2008; Osako et al., 1979). During the injection procedure Preyer's acoustic startle reflexes were tested daily and only observed at the early injection period (∼ P14–16; Jero et al., 2001). The deafening effect was confirmed by testing the auditory brainstem response (ABR) to broadband click stimuli and pure tones before the surgery. For a subset of the animals the absence of hair cells in the cochlea was also histologically confirmed.

For ABR recordings all animals were anaesthetized using an I.P. injection of a mixture of ketamine (80 mg/kg) and xylazine (12 mg/kg). Acoustic stimuli were delivered to each ear separately through hollow ear bars. Click stimuli consisted of 500 μs white noise pulses presented at 23 Hz, with 400 presentations per intensity level from 0 to 95 dB sound pressure level (SPL), in 10 dB steps. Pure-tone stimuli were presented at frequencies of 3, 6, 9 and 12 kHz, with intensities ranging from 30 to 100 dB SPL, in 10 dB steps. For details see Rosskothen-Kuhl et al. (2021). Rats of the AD + acute CI cohort were not systemically deafened prior to implantation and recording. Instead conductive hearing loss was induced during the surgical approach to the cochlea, which required removal of the tympanic membrane and ossicular chain to open the cochlea using cochleostomy (Fig. 1).

### Cochlear implantation and measurement of electrical ABR

Rats of the two acutely implanted cohorts (AD + acute CI and ND + acute CI) were bilaterally supplied with CIs directly prior to *in vivo* electrophysiological recordings. In contrast animals of the chronically implanted cohort (ND + chronic CI) were bilaterally supplied with CIs in young adulthood (P70–98), followed by 2–9 months of two-alternative forced choice (2-AFC) behavioural training and testing, as described previously in Rosskothen-Kuhl et al. (2021), Buck et al. (2023), Buchholz et al. (2024), Schnupp et al. (2025) and briefly below (Fig. 1).

CI implantation was performed as described in previous studies from our group (Rosskothen et al., 2008; Rosskothen-Kuhl et al., 2021; Buck et al., 2023). Briefly rats were anaesthetized using an I.P. injection of a mixture of ketamine (80 mg/kg) and xylazine (12 mg/kg). In addition at the beginning of the surgery the rats were given carprofen (4–5 mg/kg) s.c. as an

analgesic and amoxicillin trihydrate (15 mg/kg) s.c. as anti-inflammatory prophylaxis. During the surgical and experimental procedures the body temperature was maintained at 38°C using a feedback-controlled heating pad. The vital signs and depth of anaesthesia of the animals were frequently checked. The animal was first positioned with the external auditory canal facing upward, and a retroauricular approach was used to expose the tympanic membrane and the auditory tuberosity. The tympanic membrane was punctured, followed by removal of the malleus and incus, exposing the cochlea. The bullar wall around the cochlea was removed using a rongeur to widen the field of view. Finally a 0.6 mm diameter drill was used to perform a cochleostomy over the middle turn. Two electrode sites from the CI electrodes, with approximately 1 mm inter-site spacing, were used with one serving as stimulation site and the other as ground. The electrode array was inserted using the cochleostomy into the middle turn of the cochlea, corresponding to the ∼8 kHz region. In the ND + chronic CI cohort CI arrays were connected to a custom head connector, as previously described in Buck et al. (2023), Buchholz et al. (2024) and Schnupp et al. (2025). The head connectors were screwed onto the skull and fixed with dental cement. To get access to the area directly above the IC for electrophysiological recordings an 8% agar block was placed on the skull. Postoperatively all rats had permanent access to water for 3 days and received carprofen (4–5 mg/kg) s.c. on the 2 days following surgery. All implanted CI animal arrays were obtained from MED-EL Medical Electronics, Innsbruck, Austria.

After CI implantation, as well as directly before recordings in the ND + chronic cohort, electrically evoked ABRs (EABRs) were measured in all cohorts, as described in Rosskothen-Kuhl et al. (2021), to confirm the effectiveness of bilateral CI stimulation. The EABR amplitudes corresponding to different current intensities were used to initially estimate the minimally effective electrical stimulus amplitude.

### 2-AFC training and testing of the chronic CI cohort

Prior to the terminal electrophysiology experiments described here the six animals of the ND + chronic CI cohort had undergone 2–9 months of training and testing in 2-AFC psychoacoustic ITD and interaural level difference (ILD) lateralization tasks. For these tasks the animals were placed in the cage of a custom-made behavioural set-up with three water spouts on one side. Licking the central water spout triggered the same binaural pulse trains used in the electrophysiological experiments described in the Stimulus design section. The animals then had to choose one of the two lateral water spouts as indicated by the stimulus ITD to obtain a water reward. Incorrect responses triggered a timeout of a few seconds. Animals were trained 5 days a week, with two training sessions per day and at least 5 h between sessions. Each training session lasted between 20 and 30 min. All six chronic CI rats of this study developed good behavioural ITD and ILD sensitivity, as shown in Buchholz et al. (2024).

### Extracellular multiunit recording

Rats were anaesthetized using a mixture of ketamine (80 mg/kg) and xylazine (12 mg/kg). For maintenance of anaesthesia during electrophysiological recordings a pump delivered an I.P. infusion of 0.9 % saline solution, ketamine (17.8 mg/kg/h) and xylazine (2.7 mg/kg/h) at a rate of 3.0 ml/h in the ND + chronic CI cohort. In both AD + acute CI and ND + acute CI cohorts the level of anaesthesia was assessed by pedal reflex (firm toe pinch) every 20 min, and anaesthetic delivery was adjusted as appropriate to maintain surgical plane of anaesthesia. Either left or right IC was accessed through a craniotomy performed anterior to lambda and lateral to the sagittal sinus in anaesthetized rats fixed in a stereotaxic frame, exposing the overlying occipital cortex. A single-shaft silicon electrode array (ATLAS Neuro-engineering, E32-50-S1-L10) was inserted dorsoventrally into the IC. The electrode comprised 32 platinum recording sites spaced at 50 μm intervals. For each penetration the tip of our electrode was initially inserted to a depth of ∼4.5 mm from the brain surface. We then searched for responses by increasing the electrode depth in small increments while presenting isolated stimulus pulses at a rate of 1 to 2 per second and ranging in intensity from 0 to 12 dB (re: 100 μA = 0 dB), depending on the animal's EABR response. The search stopped when stimulus-evoked neural responses were consistently detected at the majority of recording sites, indicated by audible neuron firing and visible action potentials. In no case was it necessary to penetrate more than 0.5 mm below the initial 4.5 mm depth. Neural signals were amplified and digitized with an INTAN head stage connected via a PZ5 digitizer (Tucker-Davis Technology (TDT), Alachua, FL) to an RZ2 Bioamp processor (TDT) to be sampled and recorded at a rate of 24.414 kHz. Rats were euthanized using either overdosage of a mixture of ketamine and xylazine I.P. (ND + chronic CI cohort) or overdosage of Dorminal 20% I.P. (AD + acute CI cohort and ND + acute CI cohort) after the last recording.

### Stimulus design

The stimulation current pulses utilized in this study were generated by a TDT IZ2MH programmable constant current stimulator with a sample rate of 24.414 kHz.

The maximum stimulus amplitudes were chosen to be approximately 5 dB (re: 100 μA = 0dB) above each animal's EABR threshold and ranged from 150 to 630 μA. The electrical stimuli were pulse trains (900 pps or 4500 pps) modulated by a Hanning window. The modulation frequencies of 5, 20, and 100 Hz resulted in stimulus durations of 10, 50, and 200 ms (Fig. 2).

All electrical intracochlear stimuli consisted of pulse trains of biphasic, anode-leading current pulses (duty cycle: 40.96 μs positive, 40.96 μs at zero, 40.96 μs negative). The generated binaural electrical pulse train contained ENV ITDs and PT ITDs, varying independently within {+80, 0, −80} μs, where negative values indicated that the left ear stimulus was leading. In total there were 54 different stimulus combinations (3 ENV ITDs × 3 PT ITDs × 2 pulse rates × 3 modulation frequencies). We presented each of the 54 stimuli 15–30 times in a pseudo-randomized order.

## Data analysis

Data processing and analysis were performed using customized code written in Python. In total 28 penetrations for the ND + acute CI cohort ($n = 8$, all from left IC), 25 penetrations for the AD + acute CI cohort ($n = 6$, 22 from left IC and 3 from right IC) and 36 penetrations for the ND + chronic CI cohort ($n = 6$, all from right IC) were recorded.

**Artifact rejection.** CI stimulation pulses result in electrical artifacts characterized by sharp peaks in the recorded neural signal with a very high amplitude, typically much higher than peak values that are associated with normal neural spiking. In this study electrical pulse train stimuli with durations of 10, 50 and 200 ms were presented, resulting in an overlap between electrical artifacts and neural responses of IC neurons. To clean the data the recorded time series were cut into epochs. The epochs were linearly detrended and band-pass filtered in the frequency domain. The high-pass filter gain was a linear ramp from 0 at 80 to 1 at 300 Hz, and the low-pass filter gain was a linear ramp from 1 at 3000 to 0 at 4000 Hz. A template subtraction approach was then used to separate the electrical artifacts from the neural responses caused by the stimulation (Hashimoto et al., 2002). For each recording site and each stimulus condition a template was generated by averaging across the 15–30 repeated presentations of each stimulus. The devices for stimulus generation operate at high precision and produce highly reproducible stimuli, but the timing of neural action potentials in the midbrain is, in comparison, much more variable from trial to trial. Consequently the contribution of neural activity to the averaged waveform is reduced by phase cancellation, whereas the contribution of the stimulus artifact is preserved. The averaged waveform is therefore expected to converge to a good estimate of the stimulus artifact, and subtracting it is expected to remove most of the artifact, but small amounts of artifact residue might remain. Additional tests were therefore carried out, first to ensure that any residual artifacts would be small relative to neural response components in the cleaned waveform, and second to ensure that any residual artifacts would not introduce any significant bias in our results.

To ensure that any residual artifacts were small relative to neural responses, we developed an algorithm that looked for features in the cleaned signal that would be indicative of residual artifact and compared their size to the same features observed during baseline recording periods where no stimulus, and therefore no artifact, was present. In particular the electrical stimulus pulses are extremely steep, transitioning from zero to peak amplitudes over only a single ~40 μs wide sample. Residual stimulus artifacts should similarly be very steep, but after band-pass filtering with a corner frequency of 3 kHz, the shortest possible peaks (or valleys) are ~0.33 ms wide, and will transition from baseline to extremum over ~0.15 ms, or, equivalently, four samples. Consequently if $A(n)$ denotes the amplitude of the filtered signal, a large value for $|A(n)-A(n+4)|$ would indicate a very sharp change in voltage which might be the result of a stimulus artifact. To make our feature more specific for residual artifacts, we wanted to exploit the fact that stimulus pulses, and therefore artifact residues, occur at very precise and known interpulse intervals, denoted here by $I$. We chose to construct a feature that would measure the 'sharpest' possible peak amplitude for three pulse intervals in a row: $|A(n)-A(n+4)+A(n+I)-A(n+I+4)+A(n+2I)-A(n+2I+4)|$. We refer to this as the 'triplet feature' or just 'triplet'. Large triplet values are diagnostic of poor artifact removal, but to operationalize this fact to identify recordings in which artifact removal was deemed insufficient, we needed to estimate the range of triplet values that is to be expected in the absence of stimuli and, therefore, artifacts. For this purpose we computed for each recording epoch the 'null' distribution of triplet values observed for all samples $n$ in the baseline period, from 50 ms after stimulus offset till the end of the epoch. We also computed the maximum triple value for the part of the epoch when stimuli were present. If this maximum triplet value exceeded the 99th percentile of the corresponding null distribution, the epoch was excluded from further analysis due to suspected high levels of residual artifacts. If more than 1/4 of the epochs failed this quality control for one stimulus combination the whole multiunit was excluded from further analysis for the respective combination. Figure 3 illustrates this method.

In addition to this approach for identifying and excluding recordings for which the template-based artifact rejection method appeared insufficient, we conducted a

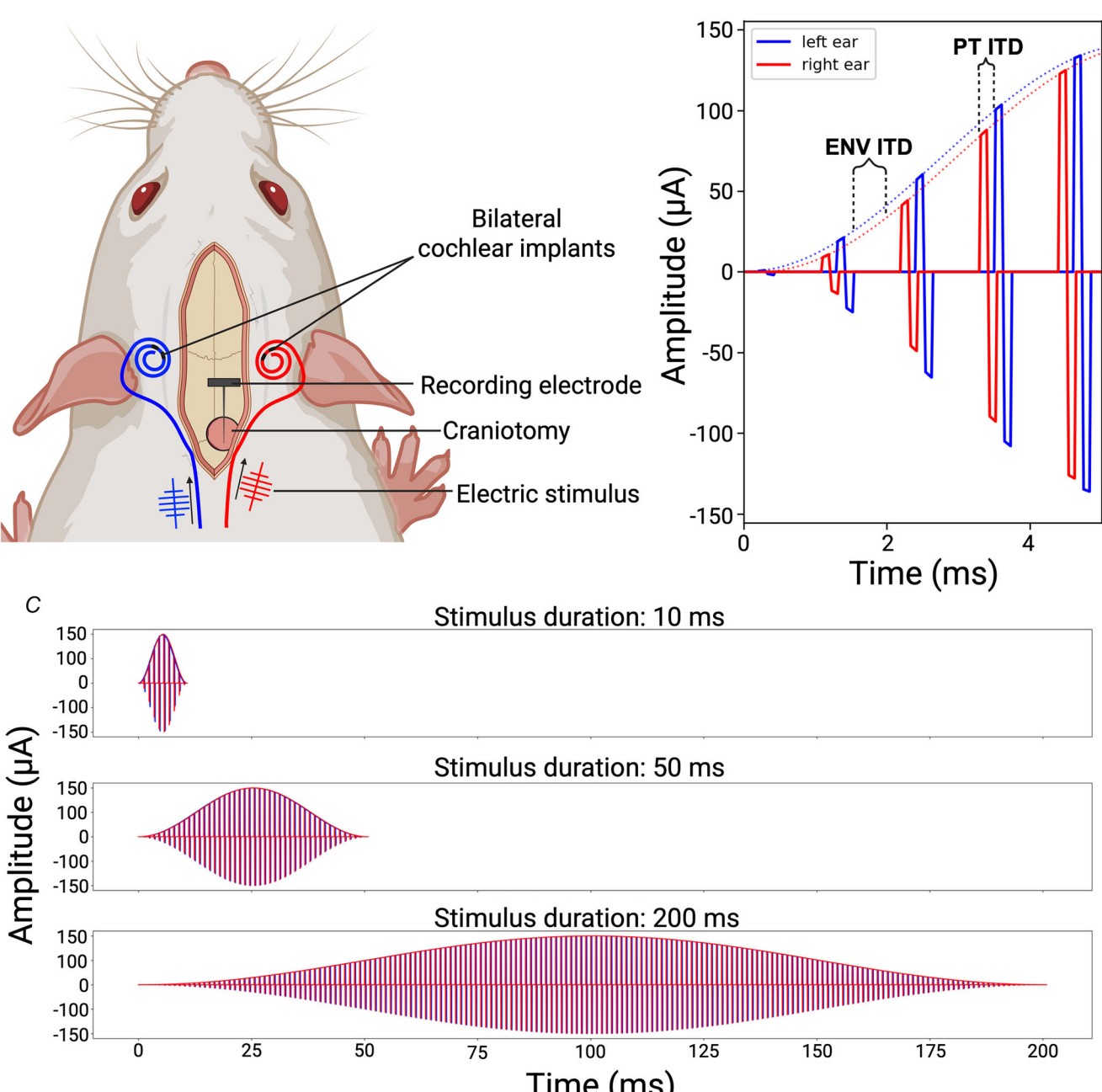

**Figure 2. Illustration of stimulus**

*A*, cartoon created using Biorender.com illustrating the experimental procedure. Rats were bilaterally implanted with cochlear implants using cochleostomy into the middle turn, and a recording electrode array was inserted into the inferior colliculus. *B*, Hanning window envelopes were used to amplitude modulate biphasic pulse trains. Envelope (ENV) and the pulse timing (PT) interaural time differences (ITDs) could be varied independently of each other. In this example the left ear (blue) ENV rises earlier than the right ear (red) ENV, illustrating a left-leading ENV ITD. Meanwhile the pulses in the left ear occur after the pulses in the right, illustrating a right-leading PT ITD. *C*, Hanning envelope durations ranging from 10 to 200 ms were chosen to approximate the envelopes that would be generated by speech sounds, from short plosives to average-duration vowels.

separate experiment to ensure that any residual artifact that might be contained in our recordings does not introduce significant bias into our results. The concern here is as follows: the electrical artifact seen by the recording electrodes results from a summation of the effects created by the left and the right CI electrodes. As a consequence the electrical artifact pulses created by stimuli with PT ITDs of -80 or +80 µs, for which the left and right ear pulses occur in succession, will be broader in time than those with zero PT ITD, for which left and right ear pluses overlap precisely. It is conceivable that the

filtering steps involved in our algorithm may not work equally well for each of the different stimulus artifact shapes created by −80, 0 or +80 µs PT ITD stimuli, respectively. In that case even small amounts of artifact residue could add different amounts of background noise to the recordings for each PT ITD condition, and if they leak through to the response strength quantification, described in the next section, they would introduce a bias into our estimates of ITD tuning strength. However any such potential, stimulus parameter-dependent 'leakage' of artifacts into the recorded responses is solely a

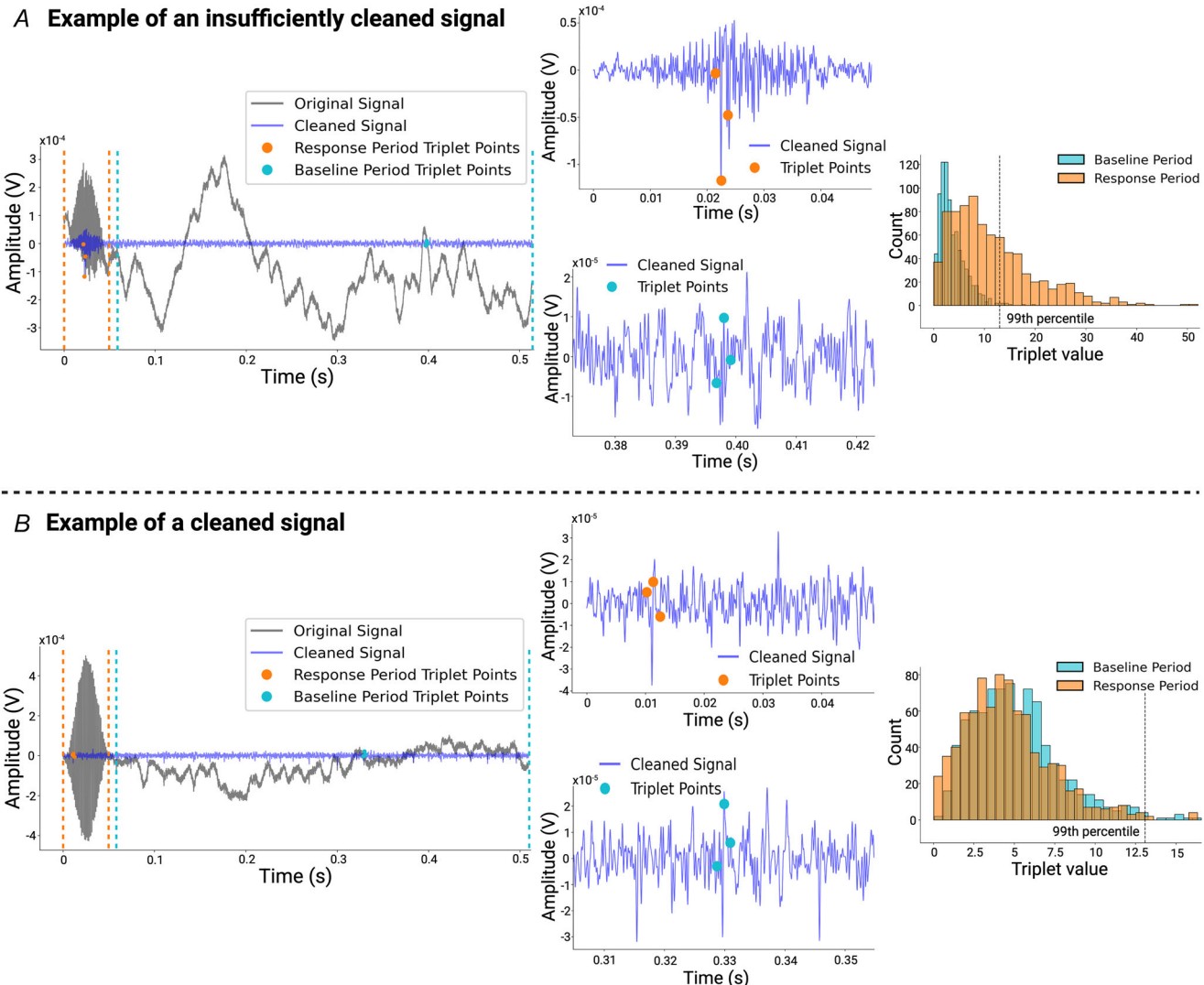

**Figure 3. Schematic overview of the artifact rejection quality control workflow**
*A*, example of a multiunit recording rejected due to residual artifacts. The 'cleaned' signal (blue) still shows leftover artifacts during stimulation time (orange dotted lines). Grey shows the raw data. Triplet points during stimulation (orange) and baseline (light blue) are labelled and magnified in the second column. The third column shows the histograms of all baseline triplet values (light blue) and response period triplet values (orange), respectively. The triplet values during stimulation time of the here shown unit exceeded the '99th percentile' criterion (black dashed line) of the 'Baseline Period' histogram. Consequently this recording was removed from further analysis. *B*, example of a multiunit that is considered artifact free and, therefore, not filtered out by the quality control. The triplet values during stimulation time do not exceed the '99th percentile' criterion in the third column.

function of the physics of the stimulation and recording equipment and of the signal processing algorithms, and it is independent of any biological mechanisms. Consequently we can use 'dummy' recordings, in which the head of an experimental animal is replaced by a block of agar and physiological saline, to verify that signals recorded by our set-up and processed by our analysis pipeline contain only negligible amounts of false-positive 'neural response variance' that is attributable to variation in stimulus parameters. We conducted such a control experiment in which 'responses' were recorded from a 1% agar in 0.9% saline block in which our 32-channel recording electrode and our binaural CI electrodes had been inserted in a configuration approximating that of our experimental animal preparation. The agar recordings were then processed and subjected to the same statistical analysis as the physiological data (see below). The results of these agar recordings can be found in Figure 4. They confirmed that after our artifact removal protocol the amount of ITD tuning attributable to leaked, residual artifact was negligible.

**Analogue multiunit activity.** Analogue multiunit activity (AMUA) was computed from the recorded extracellular voltage traces, as previously described in Kayser et al. (2007) and Schnupp et al. (2015). This method quantifies neural activity by analysing the amplitude envelope of the electrode signal in the frequency range that contains action potentials (Kayser et al., 2007; Schnupp et al., 2015). The original AMUA calculation involves

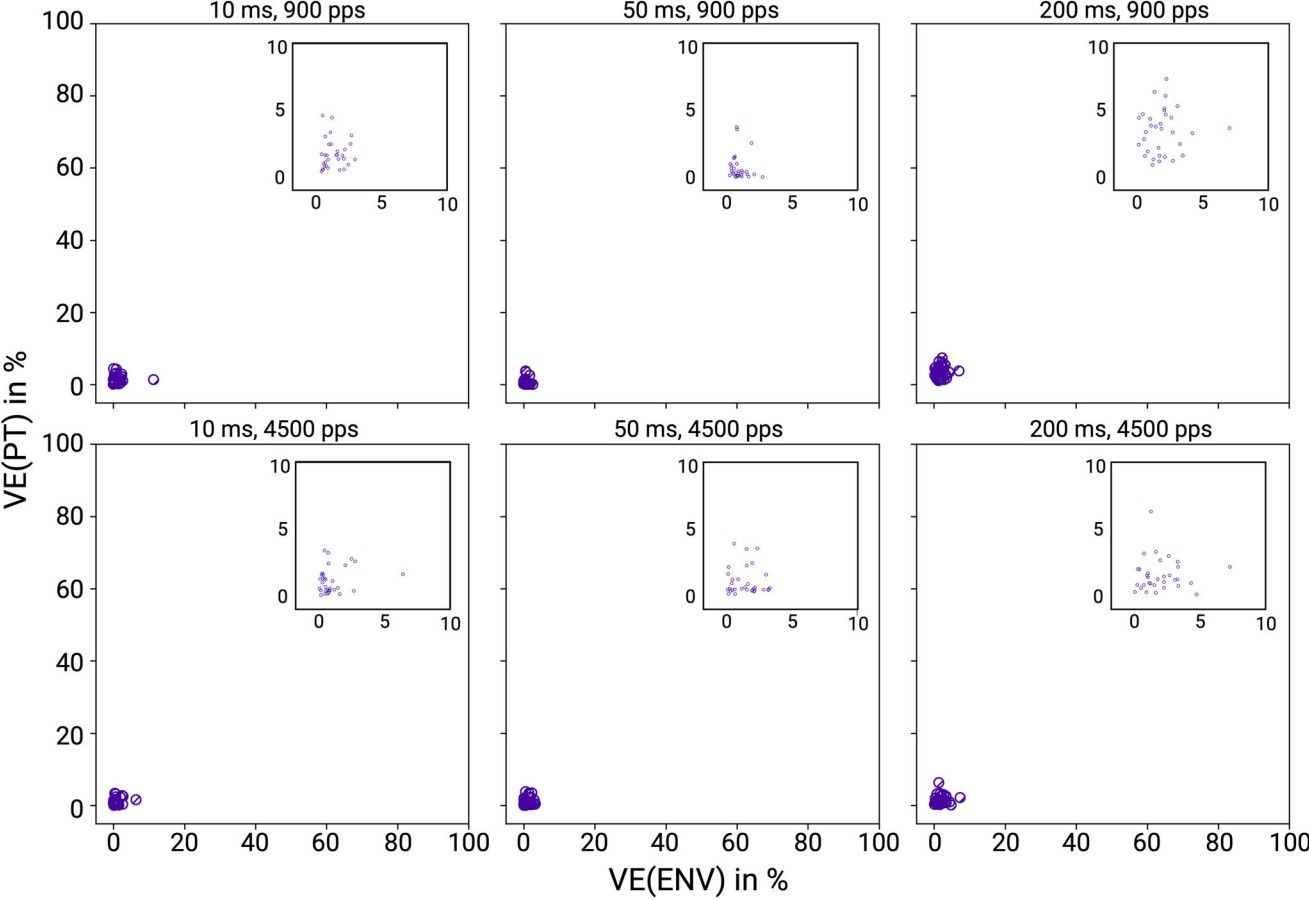

**Figure 4. Scatter plots of variance explained values from an in vitro control experiment**
Variance explained (VE) values for pulse timing ITD (PT ITD) and envelope ITD (ENV ITD) from a block of saline agar into which two cochlear implant (CI) electrodes and a 32-channel silicon electrode were inserted to mimic the in vivo measurements in CI rats. Recordings were performed just as for the experimental animals, including all three envelope durations (from left to right: 10, 50, 200 ms) and at two pulse rates 900 pps (upper row) and 4500 pps (lower row), and the agar control data were analysed and presented just as for Figure 6. Each dot shows the VE values computed for one of the 32 recording channels. Values on the *Y*-axis indicate the percentage of VE explained by PT ITD. *X*-axis indicates the percentage of VE explained by ENV ITD. As expected the VE values in this control experiment were close to zero, indicating that residual stimulation artifacts make at best only negligible contributions to the often-sizeable VE PT values observed in the AMUA responses taken from our CI rats. VE, as well as AMUA, are described in the following two paragraphs.

band-pass filtering the artifact-rejected neural signals using a fourth-order Butterworth filter (0.36 kHz). The absolute value of the filtered signal was then obtained, and a subsequent low-pass filter was applied at 6 kHz. However because our artifact rejection procedure already included a more stringent band-pass filtering (0.3–3 kHz), we simplified the AMUA calculation to include only the absolute value computation and the 6 kHz low-pass filtering steps. The resulting AMUA trace served as a measure of local multiunit firing rates that is usually less noisy than multiunit activity measures based on threshold crossings (see figure 1 in Schnupp et al., 2015). To quantify the strength of neural responses we calculated the mean AMUA response amplitude over the duration of a stimulus pulse train.

**Statistical analysis.** For each multiunit we calculated the proportion of the trial-by-trial variance in multiunit response amplitude accounted for (explained by) changes in stimulus PT ITD or ENV ITD, respectively, using an approach based on a traditional analysis of variance (ANOVA). This variance decomposition approach treats the variability in individual responses as the sum of the variability attributable to a known 'explanatory variable' or 'factor' (here, stimulus ITD) plus an unexplained, residual variability (here, trial-to-trial response variance in neural responses under repeat presentations of the same stimulus). After standard practice variability is quantified as a 'sum of squares'. If $V_{total}$, $V_{factor}$ and $V_{residual}$ denote the total sum of squares, the sum of squares for the explanatory factor and the residual sum of squares, respectively, then $V_{total} = V_{factor} + V_{residual}$, and the VE ratio can be computed simply as $VE = V_{factor}/V_{total}$ or, equivalently, $VE = 1 - (V_{residual}/V_{total})$. If we denote the different values the explanatory factor can take by $j$, the repeated observations for any one given factor by $i$, the individual observed responses by $y_{i,j}$, the mean response for a given factor by $\bar{y}_j$ and the mean response over all observations by $\bar{y}$, then:

$$VE = 1 - \left( \sum_j \sum_i \left( y_{i,j} - \bar{y}_j \right)^2 / \sum_{i,j} \left( y_{i,j} - \bar{y} \right)^2 \right)$$

In practice VE values were computed separately and independently for each of the six combinations of click rates and durations (two click rates × three durations), using the python library function statsmodels.api.stats.anova_lm(). Within each group VE was calculated separately for PT ITD and ENV ITD effects.

Comparisons between stimulus conditions are complicated by the fact that individual multiunits recorded within any one animal cannot be considered independent observations for the purposes of inferential

statistics, and that comparisons between stimulus types (PT *vs.* ENV) are 'repeated measures' multiunits, which are 'nested' in multielectrode penetrations, which, in turn, are 'nested' in individual animals. To compare sensitivities to ENV and PT, respectively, we therefore first computed the median VE values for each stimulus condition for each animal. This pooling across multiunits reduces statistical power but eliminates the difficult problem of having to adjust for the lack of independence and nesting multiple recordings from any one animal. We then analysed the median VE values using simple generalized estimating equations (GEE) models using the Python function statsmodel.formula.api.gee (see shared code for details). GEE models are extensions of ANOVA models that are well suited to deal with the fact that our dependent variable (median VE) is a strictly positive, right-skewed scalar value, which is better modelled with a Gamma than a normal distribution. The primary fixed effect was 'ITD type' (either ENV or PT), and the repeated measurements structure was accounted for. The model tested the null hypothesis of no difference in VE between ENV and PT, and statistical significance assessed via Wald tests derived from the covariance matrix. The model was used for pooled data across all cohorts and conditions and as a *post hoc* analysis of the individual conditions, divided into pulse rate and pulse duration for each group (see Fig. 7).

## Results

Twenty biCI animals were used to investigate the neurophysiological sensitivity to PT ITD and ENV ITD in the IC using three different cohorts varying in hearing experience (Fig. 1). The total number of sampled multiunits across the three cohorts was as follows: (1) 789 multiunits from 25 recording sites in AD + acute CI ($n = 6$); (2) 885 multiunits from 32 recording sites in ND + acute CI ($n = 8$); (3) 1139 multiunits from 46 recoding sites in ND + chronic CI ($n = 6$). The total number of multiunits, as well as the number of artifact-free multiunits confirmed by quality control included from each cohort for the different stimulus combinations, is given in Table 1.

### IC multiunit activity in dependence of PT and ENV ITD

To calculate the overall neural strength of our signal we used AMUA calculation as previously described in Kayser et al. (2007) and Schnupp et al. (2015). Figure 5 shows AMUA traces recorded from six illustrative multiunit examples. The top row beneath each multiunit shows how responses vary if PT ITD is held constant at 0 μs and ENV ITD is varied (−80, 0, +80 μs), whereas the bottom row shows the converse conditions, with

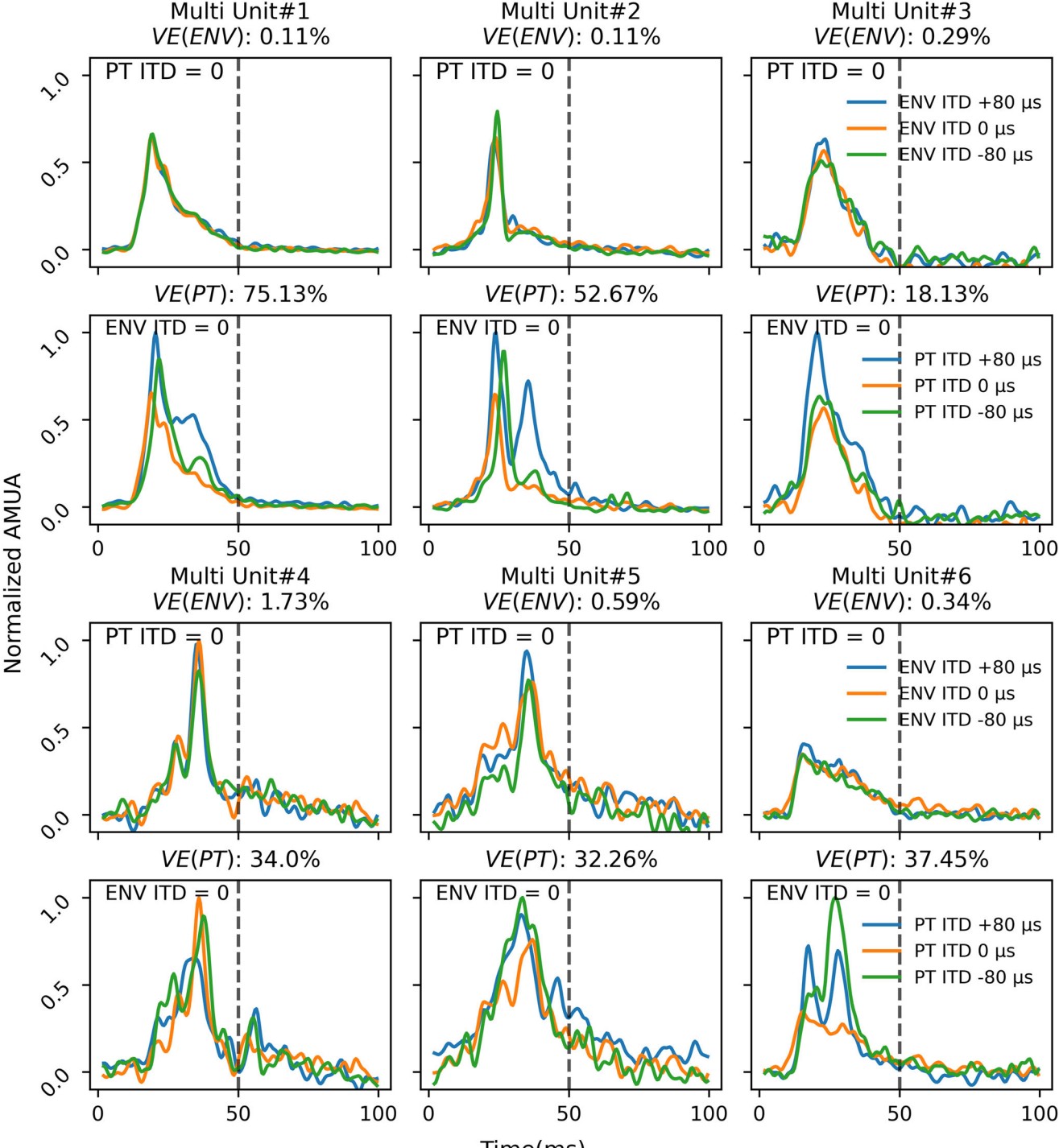

**Figure 5. Sample AMUA responses**
Samples of analogue multiunit activity (AMUA) responses from six different multiunits showing the effect of varying ENV ITD while holding PT ITD constant at 0 μS (top row beneath each multiunit) or varying PT ITD while holding ENV ITD constant at 0 μS (bottom row beneath each unit). The samples show responses to stimuli with a 50 ms duration (dotted vertical line). Each coloured line shows the averaged evoked response over all stimulus presentations, with the ENV ITD and PT ITD parameters indicated by the labels in the legends. All examples shown here were recorded at 900 pps.

**Table 1. Number of used multiunits for the analysis after artifact rejection and quality control for leftover artifacts**

| | Total | 10 ms | | 50 ms | | 200 ms | |
| --- | --- | --- | --- | --- | --- | --- | --- |
| | | 900 pps | 4500 pps | 900 pps | 4500 pps | 900 pps | 4500 pps |
| AD + acute CI | 789 | 703 | 558 | 679 | 597 | 677 | 592 |
| ND + acute CI | 885 | 681 | 760 | 654 | 711 | 706 | 718 |
| ND + chronic CI | 1139 | 991 | 935 | 919 | 620 | 1009 | 555 |

ENV ITD held constant at 0 μs while PT ITDs (−80, 0, +80 μs) vary. AMUA traces shown in Figure 5 were averaged across stimulus repetitions and normalized to their respective maximum amplitudes for comparison. The examples shown illustrate that the evoked responses change very little if ENV ITD changes, but the changes in PT ITD can induce clear changes in response shape and amplitude. Note that the examples of multiunits were selected to cover the ranges of VE(ENV) and VE(PT) that are later shown in the figures. We will examine the distributions of observed VE(PT) and VE(ENV) values further below.

### Variance-explained calculation exposes substantially greater sensitivity to PT ITD than to ENV ITD

VE analysis characterized the sensitivity to PT ITD and ENV ITD across multiunit recordings in IC. Figure 6 presents scatter plots comparing the proportion of VE by PT ITDs *versus* ENV ITDs across different stimulus durations (10, 50, 200 ms) at 900 pps and for each experimental cohort. Each individual point represents one multiunit, with colour denoting different experimental subjects.

The distribution of VE(PT) values demonstrated considerable spread along the vertical axis, with maximum values frequently exceeding 75% of total response variance, indicating robust PT ITD tuning in numerous multiunit recordings. In contrast VE(ENV) values demonstrated a notably constrained distribution proximal to zero, rarely exceeding 10% of total response variance. This marked difference in VE values indicates that a majority of multiunits exhibited substantially greater sensitivity to PT ITD compared to ENV ITD.

In addition to the 900 pps data shown in Fig. 6, Fig. 7 allows a direct comparison of the neurophysiological ITD sensitivity at stimulation rates of 900 and 4500 pps (indicated by the pale yellow background). VE values for ENV and PT ITD presentation at different stimulus durations are shown for each multiunit, with the three experimental cohorts represented in separate columns. Although the same general trends are observed as shown in Fig. 6, the box plot representation provides a clear depiction of data distribution and facilitates the identification of outliers. Across all conditions under the same stimulus condition the median VE is (orange lines in Fig. 7) is consistently higher for PT than for ENV ITD cues, and the spreading of the VE(PT) is always wider than for VE(ENV). The GEE analysis (see Methods section) over all conditions and cohorts revealed a significant effect of stimulus type on VE ($P < 0.001$), with PT producing approximately 10 times higher VE values than ENV stimuli. Individual GEE analysis comparing VE(PT) against VE(ENV) for each cohort and each pulse rate can be found on the top right corner of each subplot in Fig. 7. Across all cohorts a higher pulse rate (4500 pps) has kept approximately the same median value for VE(PT) across all stimulus durations. Accordingly no improvement in ENV sensitivity was observed in either the median values, the upper range or the overall data distribution, with no notable changes across all conditions.

## Discussion

The key finding of this study is that ITDs presented in the pulse timing of an electric stimulus elicited greater changes in neural activity in the IC than ITDs presented in the pulse train envelope. The CI stimuli we used evoked robust responses, but changing stimulus PT ITD routinely led to much greater changes in the neural responses than varying ENV ITD (compare example AMUA traces in Fig. 5). Quantifying these observations on the basis of VE calculation revealed that a high number of multiunits had substantial PT ITD sensitivity, whereas sensitivity to ENV ITD was comparatively very small and significantly less over all cohorts and conditions (Figs 6 and 7). The median VE values for PT ITD were significantly larger than those for ENV ITD regardless of stimulus type or hearing experience (see *P*-values in Fig. 7). The timing of individual pulses in the CI stimulation therefore has a much larger effect on neuronal IC responses than the timing of pulse train envelopes, a fact largely ignored in current clinical CI processing strategies, with exception of the low-frequency channels of the MED-EL FS4 strategy. Our results indicate that limited efforts made to encode auditory information in the timing of individual stimulus pulses in CI patients can result in substantially poorer sensitivity for ITD cues.

## Comparison to behavioural literature

In acoustic hearing FS and ENV ITD cues are normally coupled to one another, given that ITDs will delay both FS and ENV features at the far ear by the same amount. Previous literature on ITD presentation on FS and ENV in normal hearing psychoacoustic experiments shows much better FS than ENV ITD sensitivity using low-frequency AM tones, with FS thresholds between 60 and 180 μs and ENV thresholds often exceeding 1000 μs (Moore et al., 2018). Bernstein and Trahiotis (2002) showed that ENV ITD sensitivity in NH listeners can be much better (as

low as ∼100–130 μs) with high-frequency modulated or transposed stimuli. However comparing ITD sensitivity directly between acoustic hearing (FS and ENV) and electrical hearing (PT and ENV) is problematic due to many fundamental differences in the mechanism of natural cochlear sound transduction and electrical CI signal transduction, respectively. For example in clinical CI stimulation ENV and PT features are not coupled and usually vary independently of one another, with ENV ITDs being informed by sound source location but PT ITDs mostly created by random temporal offsets between

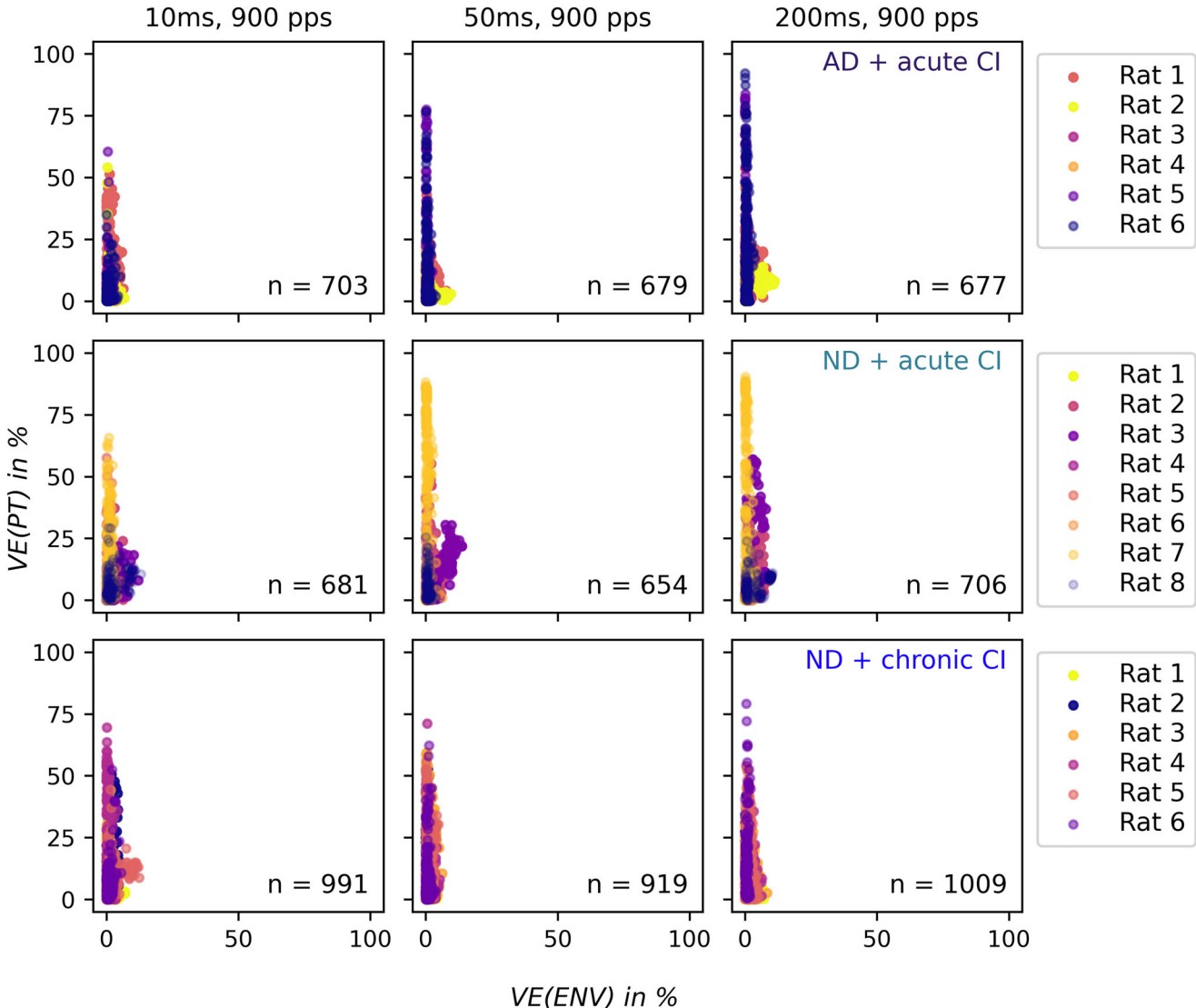

**Figure 6. Scatter plots of variance explained values across cohorts**
Increased variance explained (VE) values for pulse timing interaural time differences (PT ITD) compared to envelope ITD (ENV ITD) for all three cohorts over all three durations (from left to right: 10, 50, 200 ms) at 900 pps. All dots from one animal are labelled in the same colour. Upper row displays VE values for PT ITDs and ENV ITDs for the AD + acute CI cohort ($n = 6$). Middle row presents VE values calculated for the ND + acute CI cohort ($n = 8$). Lower row shows VE values for the ND + chronic CI cohort ($n = 6$). Values on the *Y*-axis indicate the percentage of VE explained by PT ITD. *X*-axis indicates the percentage of VE explained by ENV ITD. The respective number of multiunits can be found in Table 1.

the left and right ear pulse train generators. Although previous literature on ENV and PT ITD presentation in CI patients (Noel and Eddington, 2013; Majdak et al., 2006 and Laback et al. (2004) found PT ITD sensitivity when testing at lower pulse rates, the ITD thresholds for PT ITD presentation at high pulse rates (around 900 pps) were high. These findings are in contrast to various behavioural studies from our groups, such as Buck et al. (2023) or Schnupp et al. (2025), where we demonstrated robust behavioural sensitivity in biCI rats to even quite small ITDs, even at relatively high pulse rates of 900 pps. However it is reasonable to assume that prelingually deaf biCI users, who are exposed to confusing or uninformative PT ITDs over many months or even years, may lose sensitivity to these cues. Although the patients studied by Noel and Eddington (2013), Majdak et al. (2006) and Laback et al. (2004) were quite heterogeneous with respect to their clinical history, prior normal hearing experience or the duration of hearing experience with clinical processors delivering misleading PT ITD cues, our

cohorts differed only with respect to whether they had normal hearing experience in early life or electric hearing experience, and we did not find that these variables had a big impact on the observed neural tuning properties, as all cohorts showed much more sensitivity to PT ITD than ENV ITD (Figs 6 and 7). Multiunits with either large or small *VE(PT)* values seem to have a strong tendency to cluster with subjects (Fig. 6), and how long the 'tail' of the *VE(PT)* distribution sampled from any one animal or cohort seems to depend in large part on how often the multielectrode penetrations happened to strike clusters of neurons with large *VE(PT)*.

All early deafened biCI animals of Schnupp et al. (2025) had shown that PT ITDs dominate their behavioural lateralization decision when tested with stimuli that are essentially identical to the ones used in the electrophysiological recordings here. Our electrophysiological observations are thus in excellent agreement with this behavioural result and suggest that PT ITD very strongly modulates both IC neural responses and psychoacoustic

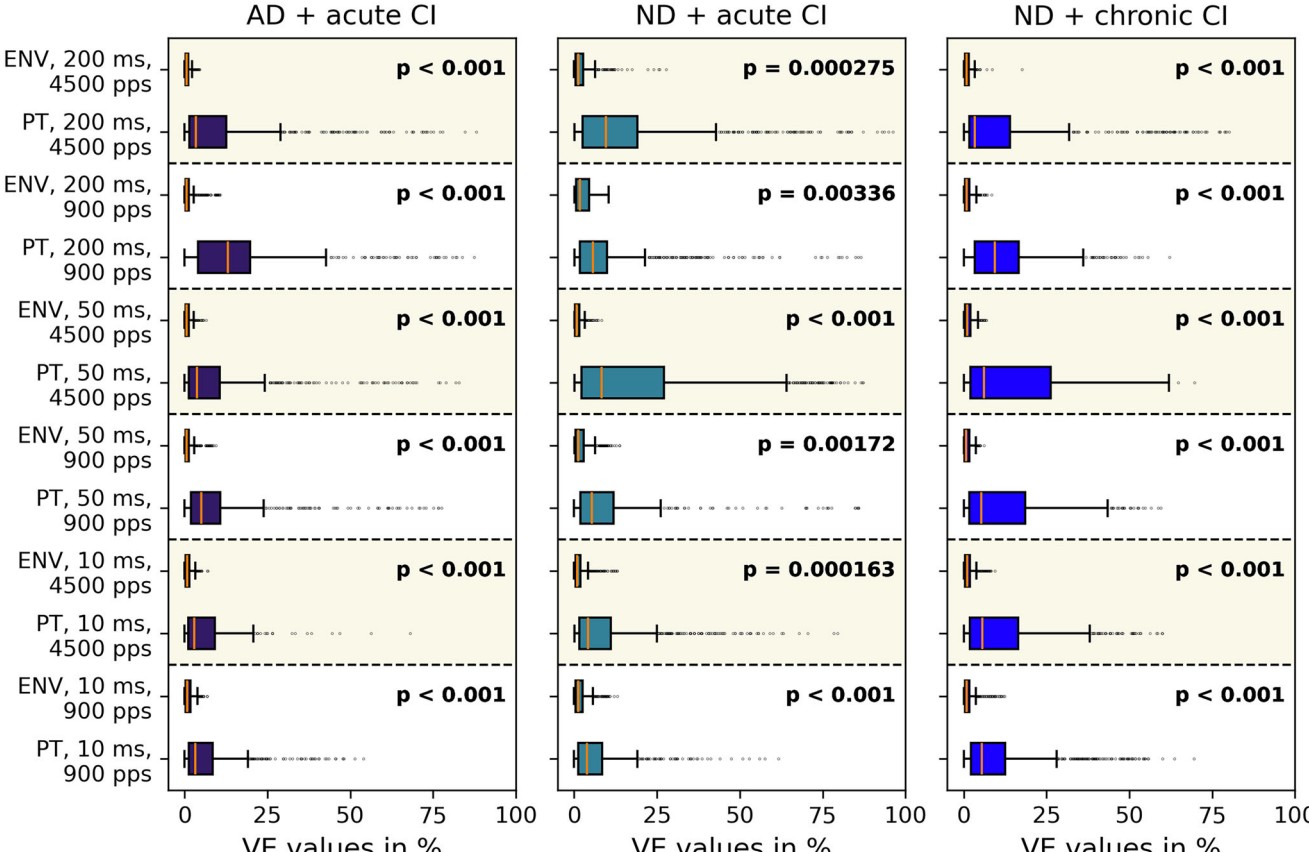

**Figure 7. Box plots of variance explained distributions for pulse timing (PT) ITD and envelope (ENV) ITD across cohorts**
Data are shown for three stimulus durations (10, 50, 200 ms; bottom to top) at 900 pps and 4500 pps stimulation rate (highlighted in pale yellow). Columns, as well as box colours, indicate the experimental cohorts (see subplot titles). Whiskers extend to 1.5× interquartile range, and data points beyond this range are plotted as outliers. *P*-values in the top right corner of each subplot display results of the GEE model for the respective condition. The respective number of multiunits can be found in Table 1.

lateralization judgments, whereas ENV ITD does so only very weakly. By investigating additional cohorts in this paper, which differed greatly in their hearing experience prior to electrophysiological recordings (one neonatally deafened with essentially no hearing experience prior to recording, another with normal hearing experience until the time of recording), we can now conclude that the dominant role of PT over ENV cues is a robust phenomenon that is not strongly influenced by auditory input during early development.

## Comparison to previous electrophysiological work in CI animals

Our results that *VE(PT)* is significantly higher in the IC than *VE(ENV)* are perhaps unsurprising when considered in the light of a previous study by Hancock et al. (2017), who studied responses of midbrain neurons to amplitude-modulated pulse trains delivered via CIs, and who observed that the IC neuron responses were frequently locked to the timing of suprathreshold pulses (shown in figure 8 in Hancock et al., 2017). It is to be expected that the timing of spikes of brainstem neurons that feed into the binaural comparison circuits in the superior olivary nuclei will similarly synchronize to the timing of individual stimulus pulses, rather than to temporal features of the pulse train ENV. In line with this Müller et al. (2023) argued that during electrical stimulation both excitatory and inhibitory brainstem neurons exhibit hypersynchronization to individual electrical pulses, which may be passed on to downstream binaural processing stages. Consequently the inputs reaching the superior olivary nuclei, and ultimately the IC, are dominated by temporally precise, pulse-synchronized activity rather than ENV-based timing cues. Their computational modelling further showed that such hyperprecise inputs maintain ITD sensitivity but within a much narrower dynamic range, consistent with a system that responds to the timing of each pulse rather than slower ENV modulations. Another study by Dynes and Delgutte (1992) also confirms this theory, as they found that electrical stimulation can produce significant phase locking of the auditory nerve well above 1000 pps.

Unlike the very sharp flanks of individual supra-threshold pulses, which can directly trigger action potentials in auditory nerve fibres, the ENV cannot be directly observed by the auditory nerve or brainstem, rather, it must be inferred ('extracted'). Consequently we would only expect ENV ITDs to be an effective binaural cue in CI stimulation if, or insofar as, mechanisms early in the auditory brainstem can extract temporal features of the ENV of a stimulus pulse train with a higher temporal resolution than the pulse rate. In the absence of such an ENV feature extraction either in the cochlear nucleus or

in the nuclei of the superior olive the timing of ENV features would inevitably be 'rounded up' to the timing of the next electric pulse triggered action potential, and PT ITDs would effectively 'overwrite' ITDs of ENV features. The significant dominance of PT ITDs over ENV ITDs documented here, which occurs at all tested parameters and all different hearing experience conditions tested, can thus be explained if we simply assume that there is only little or no 'envelope extraction filtering' in the early stages of the CI-stimulated auditory pathway, up to and including the superior olivary nuclei. Note that this situation is fundamentally different from that of normal, hair-cell based, acoustic hearing, given that the cell membrane of inner hair cells can act like an ENV extraction filter for high-frequency sound waves (Palmer & Russell, 1986). In a normally hearing auditory nerve action potential distributions can therefore track temporal ENV features of high-frequency sounds smoothly and continuously, unlike in a CI-stimulated auditory nerve, where the timing of action potentials is 'forced' onto the 'temporal grid' laid down by the timing of the electrical pulses. These fundamental differences in temporal feature encoding imply that studies of ENV ITD sensitivity in normal, acoustic hearing can be poor predictors for what to expect in the CI-stimulated case.

For example Greenberg et al. (2017) have shown in guinea pig IC that acoustic stimuli with rapid ('ramped') ENV onsets elicited much stronger ITD sensitivity than those with slow onsets ('ramped'). The best neural just noticeable difference (JND) for a steep 'ramped' envelope was ∼91 μs *versus* ∼407 μs for a slow 'ramping' ENV. This is consistent with psychophysical data from Noel and Eddington (2013), as well as Klein-Hennig et al. (2011), showing that fast onsets and long interpulse phases result in the lowest ITD thresholds. The main takeaway from these studies is that the auditory system is most sensitive to ENV ITDs when the ENV has sharp rises and well-separated pulses (Greenberg et al., 2017; Klein-Hennig et al., 2011; Noel & Eddington, 2013). In contrast our results show that for pulsatile electric stimuli delivered at a range of different pulse rates ENV ITD sensitivity remains uniformly poor irrespective of the slope of ENV onsets (Fig. 7), as we would expect if we assume that the 'rounding to the timing of the first supra-threshold pulse' that we just described is indeed a major limitation of temporal processing of CI pulse-train stimuli.

Previous studies on congenitally deaf cats with biCIs or CI-stimulated rabbits have reported relatively poor ITD tuning for IC neurons tested with ITDs at high pulse rates (Chung et al., 2016; Day et al., 2012; Hancock et al., 2010). However several design choices, including use of coarse ITD sampling with the large majority of ITDs sampled outside the physiological range, or the exclusion of onset responses in the study design, may have led

these studies to underestimate the ITD sensitivity that may be achievable with CI stimulation under more optimal conditions, and we have previously reported that IC neurons of ND biCI rats that were tested predominantly in the physiological range exhibited ample innate sensitivity even to very small ITDs (Buck et al., 2021).

The first studies to show this remarkably good sensitivity to CI ITD stimuli even in the absence of hearing experience (Buck et al. 2021, 2023; Rosskothen-Kuhl et al. 2021) used stimuli where ENV and PT ITDs covaried, as they normally would in acoustic hearing. However this experimental design does not shed any light on whether neural ITD sensitivity is triggered by one of these or a combination of both. The results presented here now provide a clear answer to this question in favour of PT ITDs. The only animal biCI study known to the authors that precisely discriminates between ITD sensitivity for PT and ENV ITD was done by Smith and Delgutte (2008). Similar to our AD + acute CI cohort cats were deafened in adulthood and 7–14 days later acutely implanted on both sides prior to electrophysiological single-unit recordings. Smith and Delgutte (2008) presented 1000 and 5000 pps pulse trains with fixed PT ITD but continuously, slowly varying ENV ITD. Smith and Delgutte concluded that many IC neurons were sensitive to ITD presentation in ENV and PT for respective frequencies. By comparing the effect of ENV ITD modulation to that of changing PT ITDs in fixed step sizes they observed a higher sensitivity to small steps in PT ITD than in ENV ITD at high pulse rates typically used in clinical settings (1000 pps). In that regard our results in CI rats of different hearing experiences are generally in good agreement with those by Smith and Delgutte (2008). One difference we note, however, is that Smith and Delgutte (2008) observed almost no neural sensitivity to FS ITDs at 5000 pps, whereas we still observe many multiunits with significant PT ITD tuning at 4500 pps (Fig. 7). At present we do not know whether these differences are due to species differences or methodological differences. In any event our results support Smith and Delgutte's overall conclusion that a coding strategy of biCIs should control precise timing of current pulses based on the fine timing of the sound source to convey ITD cues successfully.

Finally we would like to draw attention to two publications that corroborate our findings that multiunits in the IC display greater PT ITD sensitivity than ENV ITD sensitivity. First, studies by Thompson et al. (2021) in ND biCI cats showed that using a research processor with a customized 'ITD-aware' stimulation strategy, thereby presenting informative PT ITDs, results in better neurophysiological ITD sensitivity compared to using CIS-based coding strategies. Second, Sunwoo et al. (2021) stimulated ND biCI rabbits with the experimental 'fundamental asynchronous stimulus timing' (FAST) coding strategy, where PT is extracted from the temporal features of incoming sounds and thereby informative. They found that experience with FAST improved ITD sensitivity in the IC compared to animals with no stimulation experience.

## Conclusion

In normal hearing ITDs serve as powerful cues for determining the source of sound. Our research demonstrated that small ITDs are only neurophysiologically relevant under biCI stimulation if they are transmitted via pulse timing rather than the envelope of the stimulus, regardless of the envelope shape, the pulse duration, the pulse rate or the hearing experience. The inadvertent and frequently contradictory PT ITDs generated by the majority of existing clinical processors using CIS-based coding strategies likely prevent the auditory system from accessing potentially valuable binaural information. With few exceptions (notably low-frequency channels of the MED-EL FS4 strategy) current clinical devices do not encode the precise temporal details of acoustic signals with the sub-millisecond accuracy required. Our results suggest that this absence of precisely timed stimulation pulses in contemporary clinical processing is likely a major contributing factor for suboptimal binaural hearing performance seen in biCI patients.

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

## Additional information

### Data availability statement

All data supporting the findings of this study with $n < 30$ are fully included in the manuscript figures. All data ($n > 30$), as well as the analysis code used to generate all the figures and statistical results included in this manuscript, can be downloaded from https://auditoryneuroscience.org/dataShare.

### Competing interests

No competing interests declared.

### Author contributions

S.F.: conceptualization, data curation, software, validation, formal analysis, writing original draft, writing – review and editing. T.F.: conceptualization, data curation, software, formal analysis, validation, writing original draft, writing – review and editing. F.P.: data curation, software. S.B.: data curation. M.Z.: data curation. N.R.-K.: conceptualization, resources, supervision, validation, formal analysis, funding acquisition, writing original draft, writing – review and editing. J.W.S.: conceptualization, resources, software, supervision, validation, formal analysis, funding acquisition, writing original draft, writing – review and editing.

### Funding

We gratefully acknowledge funding support from the Hong Kong General Research Fund (grants 11100219, 11101020 and 11103524), the Hong Kong Health and Medical Research Fund

(grant 07181406), MED-EL Medical Electronics, Innsbruck, Austria (Research Agreement PVFR2019/2), the Research Commission of the Medical Faculty of the Medical Centre at the University of Freiburg and the charity 'Taube Kinder lernen hören e. V'.

## Keywords

binaural hearing, cochlear implants, electrophysiology, inferior colliculus, interaural time differences

## Acknowledgements

We thank B. Castellaro for surgical support. We thank E. Becker for support with training the animals.

## Supporting information

Additional supporting information can be found online in the Supporting Information section at the end of the HTML view of the article. Supporting information files available:

**Peer Review History**

## Translational perspective

Cochlear implants (CIs) are the world's most successful neuroprosthetic and help to restore hearing in individuals with severe-to-profound hearing loss. Although CIs are highly effective for understanding speech in quiet settings, many bilateral CI users struggle with localizing sounds, a skill that relies heavily on detecting tiny differences in the timing of sounds arriving at the two ears, known as interaural time differences (ITDs). The majority of current clinical CI processors do not preserve informative ITD cues in the precise timing of individual electrical pulses (pulse timing), leaving users reliant on the less salient ITD information available in the slowly varying amplitude modulation pattern (envelope). We hypothesized that neurons in the inferior colliculus, an auditory midbrain region linked to sound localization, exhibit greater sensitivity to pulse timing ITDs than to envelope ITDs. By recording neural responses in rats fitted with bilateral CIs we found that inferior colliculus neurons are indeed far more sensitive to pulse timing ITDs than to envelope ITDs across a range of stimulation conditions, regardless of prior hearing experience. These neurophysiological findings indicate that the poor spatial hearing observed in CI users may be partly due to the lack of providing informative pulse timing ITD cues with current clinical processors. Delivering informative pulse timing ITD cues through CI processors could substantially improve sound localization for CI patients and thus highlights the importance of fine-structure coding strategies for electric hearing with bilateral CIs.

