## [Peer Review History · The Journal of Physiology]

Neurophysiological Sensitivity to Envelope and Pulse Timing Interaural Time Difference in Cochlear Implanted Rats with Different Hearing Experiences

Shiyi FANG, Tim Fleiner, Fei Peng, Sarah Buchholz, Muhammad Zeeshan, Nicole Rosskothén-Kuhl, and Jan Wilbert Schnupp

DOI: 10.1113/JP290143

Corresponding author(s): Jan Schnupp (jschnupp@cuhk.edu.hk)

The following individual(s) involved in review of this submission have agreed to reveal their identity: John Middlebrooks (Referee #2)

Review Timeline:

Submission Date:	19-Sep-2025
Editorial Decision:	31-Oct-2025
Revision Received:	04-Feb-2026
Editorial Decision:	25-Feb-2026
Revision Received:	16-Mar-2026
Accepted:	27-Mar-2026

Senior Editor: Nathan Schoppa

Reviewing Editor: Conny Kopp-Scheinflug

Transaction Report:

Re: JP-RP-2025-290143 "Neurophysiological Sensitivity to Envelope and Pulse Timing Interaural Time Difference in Cochlear Implanted Rats with Different Hearing Experiences" by Shiyi FANG, Tim Fleiner, Fei Peng, Sarah Buchholz, Muhammad Zeeshan, Nicole Rosskothén-Kuhl, and Jan Wilbert Schnupp

Dear Dr Schnupp,

Thank you for submitting your manuscript to The Journal of Physiology. It has been assessed by a Reviewing Editor and by 2 expert referees and we are pleased to tell you that it is potentially acceptable for publication following satisfactory major revision.

Please address all the points raised and incorporate all requested revisions or explain in your Response to Referees why a change has not been made. We hope you will find the comments helpful and that you will be able to return your revised manuscript within 2 months. If your article is NOT for a Special Issue, you may have 9 months to revise. If you require an extension, please contact journal staff: jp@physoc.org. Please note that this letter does not constitute a guarantee for acceptance of your revised manuscript.

REVISION CHECKLIST:

We look forward to receiving your revised submission.

Yours sincerely,

Nathan Schoppa
Senior Editor
The Journal of Physiology

REQUIRED ITEMS

- Include a Key Points list in the article itself, before the Abstract.

- Author photo and profile. First or joint first authors are asked to provide a short biography (no more than 100 words for one author or 150 words in total for joint first authors) and a portrait photograph. These should be uploaded and clearly labelled together in a Word document with the revised version of the manuscript. See Information for Authors for further details.

- The contact information for the person responsible for 'Research Governance' at your institution needs to be provided. This includes their name and an institutional email address. Please ensure the contact is not an author on this paper and provide an alternate contact if necessary, or confirm in the submission form that the author whose email was provided has sole responsibility for research governance. This is the person who is responsible for regulations, principles and standards of good practice in research carried out at the institution, for instance the ethical treatment of animals, the keeping of proper experimental records or the reporting of results.

- You must start the Methods section with a paragraph headed Ethical approval (https://jp.msubmit.net/cgi-bin/main.plex?form_type=display_requirements#methods).

Research must comply with The Journal's policies regarding animal experiments (<https://physoc.onlinelibrary.wiley.com/hub/animal-experiments>) and adherence to these policies must be stated in the manuscript.

Authors should confirm in their Methods section that their experiments were carried out according to the guidelines laid down by their institution's animal welfare committee, including an ethics approval reference number. The Methods section must contain a statement about access to food, water and housing, details of the anaesthetic regime: anaesthetic used, dose and route of administration, and method of killing the experimental animals.

- Please upload separate high-quality figure files via the submission form.

- Papers must comply with the Statistics Policy: https://jp.msubmit.net/cgi-bin/main.plex?form_type=display_requirements#statistics.

In summary:

- If $n \leq 30$, all data points must be plotted in the figure in a way that reveals their range and distribution. A bar graph with data points overlaid, a box and whisker plot or a violin plot (preferably with data points included) are acceptable formats.

- If $n > 30$, then the entire raw dataset must be made available either as supporting information, or hosted on a not-for-profit repository, e.g. FigShare, with access details provided in the manuscript.

- 'n' clearly defined (e.g. x cells from y slices in z animals) in the Methods. Authors should be mindful of pseudoreplication.

- All relevant 'n' values must be clearly stated in the main text, figures and tables.

- The most appropriate summary statistic (e.g. mean or median and standard deviation) must be used. Standard Error of the Mean (SEM) alone is not permitted.

- Exact p values must be stated. Authors must not use 'greater than' or 'less than'. Exact p values must be stated to three significant figures even when 'no statistical significance' is claimed.

- Please include an Abstract Figure file, as well as the Figure Legend text within the main article file. The Abstract Figure is a piece of artwork designed to give readers an immediate understanding of the research and should summarise the main conclusions. If possible, the image should be easily 'readable' from left to right or top to bottom. It should show the physiological relevance of the manuscript so readers can assess the importance and content of its findings. Abstract Figures should not merely recapitulate other figures in the manuscript. Please try to keep the diagram as simple as possible and without superfluous information that may distract from the main conclusion(s). Abstract Figures must be provided by authors no later than the revised manuscript stage and should be uploaded as a separate file during online submission labelled as File Type 'Abstract Figure'. Please also ensure that you include the figure legend in the main article file. All Abstract Figures should be created using BioRender. Authors should use The Journal's premium BioRender account to export high-resolution images. Details on how to use and access the premium account are included as part of this email.

- Please include a full title page as part of your main article (Word) file, which should contain the following: title, authors, affiliations, corresponding author name and contact details, keywords, and running title.

EDITOR COMMENTS

Reviewing Editor: Comments to the Author:

Your manuscript has been reviewed by two experts in the field. Both agreed that the manuscript contains important information that will further the field. They also both raised concerns regarding artifact contamination and stimulus design (to assess PT over ENV on ITD sensitivity) that need detailed revision. Please tend to the detailed comments of the reviewers.

Senior Editor:

Comments for Authors to ensure the paper complies with the Statistics Policy:

I have several substantial concerns with the presented data around statistics, which are outlined in detail in the SE comments to the authors.

Comments to the Author:

Thank you for submitting your study to Journal of Physiology (JP). Your manuscript has been reviewed by two expert referees, who felt that the work is addressing an interesting question and that the results, if convincingly demonstrated, would be quite significant. They however raised a number of concerns that would need to be addressed in a revised manuscript, the most important of which are around potential artifact contamination and the stimulus design. I also have some concerns around statistical analyses (see below). My concerns will need to be thoroughly addressed with new analyses and/or an explanation in the study that justifies the methods used. A revised manuscript may potentially need to be looked at by JP's statistics editor, but I wanted to provide the authors an opportunity to address my concerns first.

Concerns around statistics:

(1) In the data in Fig. 4, it appears that the authors group the measurements across all animals tested and draw conclusions based on the 95th percentile of all the grouped data. The significance of the results are however difficult to evaluate when

the data are analyzed this way, and no precise conclusions can be made in the absence of statistical tests. Perhaps as a result of their method of analysis, the authors use "trends" statements to describe the data in Fig. 4. It would be better if the authors could analyze measurements obtained in each animal individually and then report a more standard statistical comparison across animals.

I also have the same concerns about how the data are presented and analyzed in Fig. 5. About these results, the authors state in the text that "Across all conditions, under the same stimulus condition, the median VE is consistently higher for PT than for ENV ITD cues, and the spreading of the VE(PT) is always wider than for VE(ENV)." Without a statistical test and a resulting p-value, one cannot conclude, for example, that the VE values for PT are higher than for ENV ITD cues.

In Fig. 6, the authors do report a permutation test to evaluate significance, but the analysis still appears to be across all units in all animals.

(2) Per journal policy, the authors should end the Methods with a paragraph that describe the statistical treatment of of all of their data.

(3) Precise p-values should be reported.

(4) The authors should follow all guidelines set by the journal around presentation of statistical analyses.

REFeree COMMENTS

Referee #1:

The present study by Fang et al. presents growing evidence that limitation in the coding of interaural time differences (ITDs) by electrical hearing with cochlear implants (CI) is in part due to missing synchronization of pulse timing (PT) information delivered by the CIs. Current strategies for delivering ITDs with bilateral CI use envelope ITDs rather than PT ITDs. The authors complement their very recent findings (Schnupp et al. 2025, PNAS, doi.org/10.1073/pnas.2416697122) of behavioral sensitivity to PT in CI implanted rats with neurophysiological recordings from inferior colliculus neurons. They tested three groups of animals with different degrees of hearing experience and found that PT, rather than envelope timing, explains the variance in IC neurons responses, independent of hearing experience. They conclude that PT is the major ITD cue employed by the neural system and suggest that PT ITD information should be considered to improve bilateral CI stimulation strategies.

Major:

1.) My main point addresses the source of the PT ITD sensitivity over the ENV ITD sensitivity, and I think the paper would benefit from a more elaborated discussion on this topic. There are several observations, which suggest that the high PT ITD sensitivity is due to an onset effect. To be more precise, an onset effect of the first pulse encoded by the auditory system.

A) Onset coding should arguably not only be transmitted by time differences of PT ITDs, but also by ENV ITDs. However, the stimulus design limits the onset information in ENV ITDs.

The design does not couple the ENV ITD to PT ITD, which effectively means that in ENV ITDs the pulse amplitudes are modulated by the envelope, but are presented synchronously. The maximal difference in the pulse amplitudes is about 5 % (at the maximal rise of the 10 ms modulation). The limited dynamic range of auditory nerve fibers to encode stimulus amplitude in electrical hearing (Zeng et al. 1998, as cited by the authors) could be a limiting factor to extract any information

transmitted by the amplitude difference, which is caused by the ENV ITD. This limited source of information may explain why ENV ITDs have very limited effects.

(This point should also be considered in the legend of Figure 2, as the effect of changing the steepness of the envelope might be very limited.)

B) The peak of the AMUA signal was used for the quantifications of the AMUA. In Figure 3 this peak reflects the onset of the AMUA, as also noted by the authors (line 354ff). Is this onset effect also observed in the other two modulation durations (10 ms and 100 ms)?

If this is the case in the AMUA in response to 100 ms duration stimulation, this strongly argues for an onset coding effect.

C) There is no periodicity in the AMUA signal presented by the authors in Figure 3. Referring to Figure 5A from Schnupp et al. 2015 (blue, left), the AMUA signal is i) capable of transferring the periodicity, and ii) limited to phase lock below 500 Hz (although differences in the stimulus might affect this). This suggests that the observed effect on PT ITD is mainly encoded at the onset of the signal. Furthermore, the authors observe a higher number of multi-units with response significance to 4500 pps stimuli, compared to 900 pps stimuli. Comparing the AMUA signals between 900 pps and 4500 pps AMUA signals could allow for a comparison if an onset coding preference is modulated by the pulse rate. Please also consider addressing this in the discussion line 533-541.

Minor:

-Lines 99ff: Consider changing "reconstruct" to "extract".

-Line 107f: The authors mention an "extensive psychoacoustic literature on ENV ITD" without citing a paper. Please provide literature.

-Section "Stimulus design": How big were the stimulus amplitudes at the maximum of the envelopes? Did the stimulus amplitudes differ between the three groups of hearing experience?

-Figure 2B: The illustration shows a stimulus, where both the envelope and PT ITD are altered. Please consider drawing each of the effects separately, as these are the major variables tested in the study. One way to improve understanding of PT ITD and ENV ITD would be to split Figure 2B into two figures, showing the effects of PT ITD and ENV ITD separately.

-Lines 309f: Did only one of the two PT ITDs have to reach significance or both?

-Lines 318f: \bar{y}_i and y_{total} are written with an underscore in the text and an overscore in the formula.

-Lines 311ff: The authors write that they "calculated the proportion of the trial-by-trial variance in multi-units response amplitude accounted for (explained by) changes in stimulus PT ITD or ENV ITD". However, in the formula (line 317) they used the mean response at the i th level (\bar{y}_i), which therefore does not account for trial-by-trial (30 trials) variance, but only accounts for the variance between the levels tested. More detail would enhance the understanding of the analysis.

-346 f, Figure 1: I do not understand to which amplitude the authors normalized, as neither of the shown traces reaches a peak of 1. How was the scaling performed?

-Figure 4, 5 & 6 show data in units "%", but use different unit labeling ("% vs "in %" vs "% Multi-units [...]". Please consider uniform unit labeling.

-The chapter "Neuronal PT ITD sensitivity dominates, while Env ITD sensitivity is near chance level" would benefit from a rationale for the analysis.

-Lines 486 - 496: Please consider discussing Müller et al. 2023 (doi.org/10.3389/fnins.2022.1021541) in this context.

Referee #2:

This report presents the remarkable observation that multi-unit responses in the rat's inferior colliculus are sensitive to 80-microsecond interaural differences (ITDs) in cochlear-implant pulse trains. The focus of the present study is on whether the sensitivity is dominated by ITD in the pulse timing or in the envelopes of the modulated pulse trains. The clear answer is that pulse timing dominates the ITD sensitivity, which accords with results of a behavioral study by this research team. One is concerned, however, that the present finding of microsecond-scale ITD sensitivity in the IC might have been corrupted by

contamination of responses by electrical artifact. The artifact contamination would have had a greater effect on pulse-timing than on envelope ITD, which could explain the observed dominance of PT over ENV on ITD sensitivity. One needs to be convinced that the main result of the study is not due to contamination by electrical artifact before this report could be published.

General Comment: The stimulation and recording design of this study make it challenging to isolate and reject the electrical artifact. The biphasic electrical pulses had durations of ~123 microsec in the 0-ITD condition and 205 microsec in the +/- 80-microsec-ITD conditions. That means that the artifact occupies up to ~18% of the inter-pulse periods of 1111 microsec for the 900-pps rate and ~92% of the 222 microsec period for the 4500-pps rate. The output rate of the 41-microsec pulses would have been exactly at the 24.4-kHz input sample rate, meaning that details of pulse shapes might have been subject to aliasing. The duration of the multi-phase artifact would have been 2 pulses longer for the 80-microsec-ITD condition than for the 0-ITD condition. That means that if any electric artifact were to get past the template subtraction and contribute to mean amplitude, the mean amplitude would be greater in the 80-microsec-ITD condition. Figure 3 gives the only examples of multi-unit amplitudes. In each of the cases in the lower row (showing PT ITD effects), amplitudes were greater for +/- 80 microsec-ITD conditions than for the 0-ITD condition. That non-monotonic ITD sensitivity fits well with an explanation based on longer artifacts for non-0-ITD conditions. In contrast, the non-monotonicity conflicts with one's expectation of ITD coding in the IC and with this research group's reports of behavioral discrimination of +80 versus -80 microsec; in 2 of the 3 PT plots, the peak AMUA values did a better job of discriminating + and - 80 microsec from 0 ITD than for discriminating +80 from - 80, which is what was required in the behavior protocol. The artifact explanation is not relevant to the upper row of the figure, in which the PT ITD was constant at 0 microsec.

One might be more convinced of the effectiveness of the artifact rejection if the authors were to illustrate examples of templates and template subtraction for 0- and 80-microsec conditions and to show us how biological responses survive template subtraction. Such an illustration might also help explain the quality-control procedure given in lines 278-290, which this reader found difficult to understand.

Specific comments:

Lines 76-77: "very limited ability" to provide PT ITD is an understatement. Present-day processors deliver no stimulus-relevant PT ITD.

Line 92: "the adjacent electrode" should be "adjacent electrodes"

Line 96: "random numbers" is not exactly correct. The PT ITD delivered by such devices would be constant at some value unrelated to the sound-source ITD.

Line 150: "permanent access" here and later is an odd usage. "constant access" would be better.

In the chronic CI cohort, how many hours of electrical stimulation and training/testing did each rat receive each day? How many days per week?

Artifact rejection: Please clarify whether or not a unique template was computed for each stimulus condition (including for each ITD).

Lines 351-353: This sentence sounds as if one is defending against a claim that the examples were chosen with a bias against VE(ENV) conditions. It would be better just to say that the examples were selected to cover the ranges of VE values shown in later figures.

Please provide an illustration or statistic to show the distributions of multi units having minimum and maximum amplitudes at -80, 0, or +80 microsec PT ITD. Alternatively, one could show box plots of normalized mean AMUA at -80, 0, and +80 microsec. One finds it odd that all of the examples had max at +80 and min at 0 ITD.

What was the pulse rate for the examples in fig. 3?

In fig. 5, in all cases for the acute CI animals, VE(PT) was higher for 4500 than for 900 pps (also shown in fig. 6). Greater ITD sensitivity at higher pulse rates is contrary to published results for normal hearing and contrary to likely physiological mechanisms. That result, however, is what one would expect if the VE(PT) were driven by electrical artifact.

Line 408: should be "sensitive IC units for each cohort". Here, you claim an observation (% of PT ITD units increases with duration) based on N=3, with 2 supporting the observation and one exception. One might soften that statement, like "% of PT ITD units generally increased".

Lines 475-476: Please remind us here that Schnupp et al. (2025) was a behavioral study.

END OF COMMENTS

The Journal of Physiology Decision on Our Manuscript: "Neurophysiological Sensitivity to Envelope and Pulse Timing Interaural Time Differences in Cochlear Implanted Rats with Different Hearing Experiences", *JP-RP-2025-290143*.

Dear Prof. Schoppa,

Dear Reviewers,

we would like to thank the editors for giving us the opportunity to submit a revised version of our manuscript entitled "Neurophysiological Sensitivity to Envelope and Pulse Timing Interaural Time Differences in Cochlear Implanted Rats with Different Hearing Experiences" In addition, we thank the reviewers for their positive reception of our work as well as their thoughtful and constructive criticisms of our original manuscript. We have submitted a detailed reply to the reviewers' comments in the "Point-by-point response" attached below. Please also find our revised manuscript (tracked changes mode), in which we have highlighted all amendments to our manuscript.

In our rebuttal and the revised manuscript, we acknowledge that we have now adapted the analysis and artifact rejection pipeline thoughtfully to optimize artifact rejection. We further expanded the statistical analysis to implement a Generalized Estimating Equations model to report precise p-values and to verify the effect size. We have revised the text throughout to improve interpretation and have now explicitly noted the origin of these data in the relevant sections. We have added all details and revised the manuscript according to the reviewers' comments.

Again, we thank the editor and the reviewers for their time and their constructive suggestions. We hope that, after considering our point-by-point responses and our edits to the manuscript, the editor and reviewers will find our paper greatly improved.

Yours sincerely,

Shiyi Fang, Tim Fleiner, Fei Peng, Sarah Buchholz, Muhammad Zeeshan, Nicole Rosskoth-Kuhl and Jan W. Schnupp

EDITOR COMMENTS

Reviewing Editor: Comments to the Author:

Your manuscript has been reviewed by two experts in the field. Both agreed that the manuscript contains important information that will further the field. They also both raised concerns regarding artifact contamination and stimulus design (to assess PT over ENV on ITD sensitivity) that need detailed revision. Please tend to the detailed comments of the reviewers.

Senior Editor:

Comments for Authors to ensure the paper complies with the Statistics Policy:

I have several substantial concerns with the presented data around statistics, which are outlined in detail in the SE comments to the authors.

Comments to the Author:

Thank you for submitting your study to Journal of Physiology (JP). Your manuscript has been reviewed by two expert referees, who felt that the work is addressing an interesting question and that the results, if convincingly demonstrated, would be quite significant. They however raised a number of concerns that would need to be addressed in a revised manuscript, the most important of which are around potential artifact contamination and the stimulus design. I also have some concerns around statistical analyses (see below). My concerns will need to be thoroughly addressed with new analyses and/or an explanation in the study that justifies the methods used. A revised manuscript may potentially need to be looked at by JP's statistics editor, but I wanted to provide the authors an opportunity to address my concerns first.

We are grateful for the opportunity to address these concerns directly. Based on the feedback from the editor and reviewers, we have re-evaluated our electrophysiological analysis, conducted additional tests to ensure that residual electrical artifacts do not bias our results, and significantly improved and simplified our statistical analysis pipeline and the description of the analysis. We have revised the results section and figures in our manuscript accordingly. For details, please see below.

Concerns around statistics:

(1) In the data in Fig. 4, it appears that the authors group the measurements across all animals tested and draw conclusions based on the 95th percentile of all the grouped data. The significance of the results are however difficult to evaluate when the data are analyzed this way, and no precise conclusions can be made in the absence of statistical tests. Perhaps as a result of their method of analysis, the authors use "trends" statements to describe the data in Fig. 4. It would be better if the authors could analyze measurements obtained in each animal individually and

then report a more standard statistical comparison across animals.

I also have the same concerns about how the data are presented and analyzed in Fig. 5. About these results, the authors state in the text that "Across all conditions, under the same stimulus condition, the median VE is consistently higher for PT than for ENV ITD cues, and the spreading of the VE(PT) is always wider than for VE(ENV)." Without a statistical test and a resulting p-value, one cannot conclude, for example, that the VE values for PT are higher than for ENV ITD cues.

The editor is correct that our original Figure 4 (now Fig. 5) did not present the data in a manner that lends itself to illustrating a standard significance test, but that was never the intention behind this figure. Rather, we were hoping the figure would allow the readers to develop an intuition for the distribution of variance explained (VE) tuning strength measures observed in our sample. The subpanels of Figure 5 show that the VE distributions for PT ITD have very long tails while the tails for ENV ITD are much shorter, and this clearly appears to be the case in all stimulus conditions and cohorts. Plotting lines highlighting the 95th centile was merely intended to make the comparison of the length of the distributional tails more obvious, but in hindsight it was a mistake because it created the wrong impression that we intended to derive any formal conclusions from these specific centile values. We have therefore now removed these 95% centile lines in the revision.

To address the reasonable request that we should add a more standard statistical test to show more formally that the very obvious differences in the VE ENV and VE PT value distributions are robust when animal-to-animal variance is also taken into account, we have now introduced tests using Generalized Estimating Equations (GEE), an extension of ANOVA which is suitable for repeated measures and data with positive skew and non-normal variance, to investigate the effects of ITD type (ENV or PT) or stimulus duration on VE values. The approach is described in the revised Methods, and the results are reported in and around the revised version of Figure 5, now Fig. 6.

In Fig. 6, the authors do report a permutation test to evaluate significance, but the analysis still appears to be across all units in all animals.

This permutation test has been removed in the revised manuscript, along with former Figure 6. The information in it was somewhat redundant. It stands to reason that, if the median VE in one condition is higher than in another condition, that the proportions of units for which the VE exceeds a significance threshold would most likely also be elevated. The GEE results just described and the new Figures 5 and 6 illustrate the very large, highly consistent and statistically very highly significant differences in the VE PT and VE ENV distributions in great detail. Incorporating additional permutation tests to count the number of units for which VE exceeds an ultimately arbitrary significance threshold now seems superfluous for the purposes of this study.

(2) Per journal policy, the authors should end the Methods with a paragraph that describe the statistical treatment of all of their data.

We would like to apologize to you for not doing that in the first place. We have now added a precise summary and a detailed description of the analysis methods used in the manuscript at the end of the Methods section.

(3) Precise p-values should be reported.

We updated the statistical methods used in the manuscript as described above and tried to report precise p-values following the journal's guidelines where they are required in the Results section. p-values of the GEE model are reported in Figure 6, top right corner of each cohort and condition, illustrated in each subplot. The GEE model over all conditions and cohorts revealed a significant effect of stimulus type on VE ($p < 0.001$), with PT stimuli producing approximately 10 times higher responses than ENV stimuli and is reported in the Results part, shortly above this Figure.

(4) The authors should follow all guidelines set by the journal around presentation of statistical analyses.

We would like to thank the editor for the comments and the reminders about the journal's guidelines. We tried our best to follow every guideline conscientiously in our revised manuscript.

REFEREE COMMENTS

Referee #1:

The present study by Fang et al. presents growing evidence that limitation in the coding of interaural time differences (ITDs) by electrical hearing with cochlear implants (CI) is in part due to missing synchronization of pulse timing (PT) information delivered by the CIs. Current strategies for delivering ITDs with bilateral CI use envelope ITDs rather than PT ITDs. The authors complement their very recent findings (Schnupp et al. 2025, PNAS, doi.org/10.1073/pnas.2416697122) of behavioral sensitivity to PT in CI implanted rats with neurophysiological recordings from inferior colliculus neurons. They tested three groups of animals with different degrees of hearing experience and found that PT, rather than envelope timing, explains the variance in IC neurons responses, independent of hearing experience. They conclude that PT is the major ITD cue employed by the neural system and suggest that PT ITD information should be considered to improve bilateral CI stimulation strategies.

Major:

1.) My main point addresses the source of the PT ITD sensitivity over the ENV ITD sensitivity, and I think the paper would benefit from a more elaborated discussion on this topic. There are several observations, which suggest that the high PT ITD sensitivity is due to an onset effect. To be more precise, an onset effect of the first pulse encoded by the auditory system.

A) Onset coding should arguably not only be transmitted by time differences of PT ITDs, but also by ENV ITDs. However, the stimulus design limits the onset information in ENV ITDs.

The design does not couple the ENV ITD to PT ITD, which effectively means that in ENV ITDs the pulse amplitudes are modulated by the envelope, but are presented synchronously. The maximal difference in the pulse amplitudes is about 5 % (at the maximal rise of the 10 ms modulation). The limited dynamic range of auditory nerve fibers to encode stimulus amplitude in electrical hearing (Zeng et al. 1998, as cited by the authors) could be a limiting factor to extract any information transmitted by the amplitude difference, which is caused by the ENV ITD. This limited source of information may explain why ENV ITDs have very limited effects.

(This point should also be considered in the legend of Figure 2, as the effect of changing the steepness of the envelope might be very limited.)

If we understand correctly, this point here is mostly concerned with the interpretation and discussion of our data, rather than a fundamental issue with our present study per se. A few conceptual issues appear to be in play here. Perhaps the most important issue is that the reviewer's comments are guided by their strong intuition that an overwhelmingly strong sensitivity to the timing of the first supra-threshold pulse (the "onset") is likely to explain why PT ITDs are a lot more effective than ENV ITDs, and they would like us to do more to interpret our results from that perspective.

This is a perspective that we have a lot of sympathy for. We too believe that the timing of the first supra-threshold pulse is bound to be important. Nevertheless, we hope the reviewer may appreciate that our ability to illustrate the importance of the first supra-threshold pulse using the present dataset is very limited and cannot lead to compelling results. The issue here is that, if we had set out from the beginning to try to demonstrate and quantify the relative importance of the first supra-threshold pulse, we would have designed the whole study very differently. But that was not the original aim that guided the design of this study. We apologize that we did perhaps make the purpose and scope of this study completely unambiguous in the original manuscript. We have now edited the final paragraph of the Introduction to remedy this shortcoming.

The reviewer is of course correct that experts who carefully look at Figure 2 should find it unsurprising that ENV ITDs are bound to be at a huge disadvantage relative to PT ITDs because the envelope timing would first have to be reconstructed by the auditory brainstem from pulse amplitudes that are, as far as we know, not well resolved in the auditory nerve. However, ever since the invention of the CIS coding strategy, almost the entire field (with only a few exceptions, such as work by MED-EL on their fine structure strategies) has nevertheless been expecting CI patients to rely entirely on this most likely very deficient envelope timing for ITD and temporal pitch discrimination tasks. The key objective of this paper is to document why that is bound to be a bad idea, and this objective guides the parameter choices of our stimuli. Our pulse trains are up to 200 ms long and have Hanning windows because that reasonably approximates the energy arc of spoken phonemes or syllables. Our pulses are delivered at constant rates of 900 pps or higher because those are typical clinical parameters. And we do not, as the reviewer correctly points out, couple PT ITDs or ENV ITDs because these two types of cues will vary independently in the stimulation received by the large majority of binaurally implanted patients who wear Cochlear, Advanced Bionics or Nurotron CI devices. We have edited the 2nd and 3rd paragraph of the introduction to indicate that this orientation along clinical practice motivated the choice of our stimulus parameters. We strongly believe that documenting IC response patterns to stimuli that resemble those used in clinical practice in important ways is a valuable thing to do, and this was our priority in the design of this study.

Unfortunately, we cannot, in a single study, orient our stimulus design along current clinical practice, while also and at the same time optimizing our stimuli to investigate how important the first supra-threshold pulse is for onset timing. And since here we decided to prioritize parameters that reflect the inputs that patients would typically experience, it is not realistic to expect the data presented here to also provide compelling insights into the importance of the first supra-threshold pulse in signalling onset timing. Thus, while we fundamentally agree with the reviewer's intuition that onset timing is bound to be important, and we are happy to acknowledge this in our revision, we nevertheless need to be mindful that the current dataset simply cannot provide compelling evidence for this hypothesis. Quantifying the relative importance of onset weighting encoded in spike timing as opposed to onset encoded in envelopes would require a quite different and sophisticated stimulus design that might combine variable pulse intervals in temporal weighting function (TWF) measurements with variable onset ramps. This would be a very interesting thing to do, but it is far outside the scope of the current study's design. We hope that the reviewer will therefore not be too disappointed that our revisions can only make sympathetic nods in the direction of the idea that onset encoding by pulse timing is likely to be important, but it cannot deliver compelling evidence that this is indeed the case.

B) The peak of the AMUA signal was used for the quantifications of the AMUA. In Figure 3 this peak reflects the onset of the AMUA, as also noted by the authors (line 354ff). Is this onset effect also observed in the other two modulation durations (10 ms and 100 ms)?

If this is the case in the AMUA in response to 100 ms duration stimulation, this strongly argues for an onset coding effect.

Here, the reviewer is reading too much into a small number of example traces which fail to reflect the full diversity of IC response profiles. It is true that in the three examples shown in the original Figure 3 (now Figure 4), the AMUA appeared to peak near the onset of the stimulus, but IC response patterns are very diverse, and not all multi-units had AMUA peaks near stimulus onset. Our revised Fig. 4 now includes a further three multi-units which include examples with peaks occurring later during the stimulus. We hope also that, in the light of the points made in the previous paragraphs of our reply, the reviewer will take our word for it that looking for evidence of onset coding in the timing of peak AMUA responses is not a fruitful enterprise. A differently designed stimulus set would be needed to undertake a compelling investigation into the importance of onset coding.

Please also note that we used the mean AMUA response over the entire stimulus pulse train duration to quantify neural response strength, not the peak amplitude. Our apologies that this was not spelled out clearly in the original manuscript. We now state this at the end of the revised “Analog multi-unit activity” subsection of the “Methods” section.

Regarding previous lines 354ff, we thank the reviewer for bringing this to our attention. While the original sentence was intended to describe the three examples displayed in the original Figure 3 (now Figure 4), we agree that it could potentially mislead readers to overly focus on onset responses. We have therefore removed this sentence from the manuscript.

C) There is no periodicity in the AMUA signal presented by the authors in Figure 3. Referring to Figure 5A from Schnupp et al. 2015 (blue, left), the AMUA signal is i) capable of transferring the periodicity, and ii) limited to phase lock below 500 Hz (although differences in the stimulus might affect this). This suggests that the observed effect on PT ITD is mainly encoded at the onset of the signal. Furthermore, the authors observe a higher number of multi-units with response significance to 4500 pps stimuli, compared to 900 pps stimuli. Comparing the AMUA signals between 900 ppS and 4500 pps AMUA signals could allow for a comparison if an onset coding preference is modulated by the pulse rate. Please also consider addressing this in the discussion line 533-541.

I am afraid we cannot fully follow the logic of this argument. As mentioned above, we are in agreement with the reviewer’s intuition that the timing of supra-threshold

pulses near the onset of the pulse train is bound to be very important, but the extent to which this may depend on pulse rates (and/or the shape of the onset ramp) is a complex issue which our stimulus set was never designed to adequately address. A priori, there is no good basis for hypothesising how the role of the first supra-threshold pulse on a stimulus train would change as pulse rates increase from 900 to 4500 pps. Bearing in mind also that IC multi-unit response shapes are a lot more diverse than the small number of examples shown in our Fig 4 (formerly Fig 3) or the example figures in Schnupp et al 2015 can convey, we have good reason to fear that any attempt to perform systematic comparisons of AMUA signal shapes at two pulse rates would be very complicated to perform while offering little prospect of yielding compelling insights into how onsets are encoded under CI stimulation. Again, we beg the reviewer's indulgence: investigating key determinants of CI stimulus onset coding would be a very interesting thing to do, but this study was simply not designed with that in mind, and repurposing our analysis after the fact in pursuit of a different goal is not going to yield compelling insights.

Minor:

-Lines 99ff: Consider changing "reconstruct" to "extract".

We have made the necessary adjustments in all relevant areas accordingly.

-Line 107f: The authors mention an "extensive psychoacoustic literature on ENV ITD" without citing a paper. Please provide literature.

We would like to thank the reviewer for that comment, and we added more literature at that point (now lines 108-110).

-Section "Stimulus design": How big were the stimulus amplitudes at the maximum of the envelopes? Did the stimulus amplitudes differ between the three groups of hearing experience?

We added a detailed description of the stimulus amplitudes for the different cohorts to the "Stimulus design" section, which reads as follows: "*The maximum stimulus amplitudes were chosen to be approximately 5 dB (re: 100 μ A = 0dB) above each animal's EABR threshold and ranged from 150 μ A to 630 μ A.*" Note that the stimulus amplitude ranges overlapped and did not differ significantly between cohorts.

-Figure 2B: The illustration shows a stimulus, where both the envelope and PT ITD are altered. Please consider drawing each of the effects separately, as these are the major variables tested in the study. One way to improve understanding of PT ITD and ENV ITD would be to split Figure 2B into two figures, showing the effects of PT ITD and ENV ITD separately.

We thank the reviewer for their thoughtful comment and constructive suggestion. We presented PT and ENV parameters together in each trial, and therefore chose to display both parameters combined in this figure. Separating them could create the misleading impression that only one parameter was presented at a time. Note that Figure 2B is not a rough schematic intended to illustrate a concept. Rather, it is an accurate graphical representation of one of the stimuli actually used in the study, drawn to scale. Furthermore, this plotting scheme was used consistently in our companion behavioral study (10.1073/pnas.2416697122), which is why we have retained it here for continuity and clarity. We hope the reviewer will accept that we would prefer to stick with our current presentation.

-Lines 309f: Did only one of the two PT ITDs have to reach significance or both?

We thank the reviewer for raising this point. Upon revision, we determined that the analysis in question was not essential and the information in it was somewhat redundant. We have therefore removed it from the manuscript, and the issue of significance criteria is no longer relevant.

-Lines 318f: y_i and y_{total} are written with an underscore in the text and an overscore in the formula.

Revised as requested.

-Lines 311ff: The authors write that they "calculated the proportion of the trial-by-trial variance in multi-units response amplitude accounted for (explained by) changes in stimulus PT ITD or ENV ITD". However, in the formula (line 317) they used the mean response at the i th level (\bar{y}_i), which therefore does not account for trial-by-trial (30 trials) variance, but only accounts for the variance between the levels tested. More detail would enhance the understanding of the analysis.

This is a perhaps somewhat tedious mathematical detail. The variance decomposition approach assumes that individual responses are the result of additive effects of a deterministic "factor" (here, ITD), plus "residual" variance that is independent of the factor, so $V_{total} = V_{factor} + V_{residual}$. "Variance explained" is then the proportion of the total variance that is attributable to the factor: $VE = V_{factor} / V_{total}$ as per our original equation 1. Now, unlike $V_{residual}$, which concerns itself with how individual responses vary around factor means, V_{factor} only concerns itself with how factor means \bar{y}_i vary around the total mean \bar{y}_{total} . When we reproduced the definition of V_{factor} in our original formula 2, using, as the reviewer correctly states "the mean response at the i_{th} level (\bar{y}_i), which therefore does not account for trial-by-trial (30 trials) variance, but only accounts for the variance between the levels tested", then this was as it should be. To appreciate this, consider that a VE of 1 (100%) must imply that all variance is accounted for by variance between levels, and every individual trial will land exactly on its level mean. Thus, formulas 1 and 2 were

correct, but, as the reviewer's comment suggests, they may not have been terribly intuitive. To arrive at a more intuitive formulation, given that $V_{\text{factor}} = V_{\text{total}} - V_{\text{residual}}$, one can entirely equivalently define and compute VE as $VE = 1 - (V_{\text{residual}} / V_{\text{total}})$. We have therefore replaced the formulas in the original manuscript and rewritten the first paragraph of the Statistical Analysis subsection of the Methods to provide a hopefully more accessible description:

"This variance decomposition approach thinks of the variability in individual responses as the sum of the variability attributable to a known "explanatory variable" or "factor" (here, stimulus ITD) plus an unexplained, residual variability (here, trial to trial response variance in neural responses under repeat presentations of the same stimulus). Following standard practice, variability is quantified as a "sum of squares". If V_{total} , V_{factor} and V_{residual} denote the total sum of squares, the sum of squares for the explanatory factor and the residual sum of squares respectively then $V_{\text{total}} = V_{\text{factor}} + V_{\text{residual}}$, and the variance explained (VE) ratio can be computed simply as $VE = V_{\text{factor}} / V_{\text{total}}$, or, equivalently, as $VE = 1 - (V_{\text{residual}} / V_{\text{total}})$. If we denote, the different values the explanatory factor can take by j , the repeated observations for any one given factor by i , the individual observed responses by $y_{i,j}$, the mean response for a given factor by \bar{y}_j , and the mean response over all observations by \bar{y} , then:

$$VE = 1 - \left(\sum_j \sum_i (y_{i,j} - \bar{y}_j)^2 / \sum_{i,j} (y_{i,j} - \bar{y})^2 \right)$$

"

We thus now provide an alternative formula for VE which yields identical results but references individual observations $y_{i,j}$ in both the numerator and the denominator, and which may therefore hopefully be easier to follow. We now also make it clear that, in practice, the VE values were computed using the standard python library function `statsmodels.api.stats.anova_lm()`

-346 f, Figure 1: I do not understand to which amplitude the authors normalized, as neither of the shown traces reaches a peak of 1. How was the scaling performed?

We believe the reviewer is referring here to our previous Figure 3 (now Fig. 4). In Fig. 4, we present six examples, each consisting of two vertically arranged sub-windows. All average AMUA traces shown in Fig. 4 were normalized to the maximum value of the average AMUA in the stimulus window of that respective example. A detailed description of the normalization has now been added to the paragraph above Fig. 4 (lines 424-425).

-Figure 4, 5 & 6 show data in units "%", but use different unit labeling ("% vs "in %" vs "% Multi-units [...]"). Please consider uniform unit labeling.

We added "in %" to the corresponding Figures (now Figs. 5-6).

-The chapter "Neuronal PT ITD sensitivity dominates, while Env ITD sensitivity is near chance level" would benefit from a rationale for the analysis.

As already mentioned in the reply to the editor above, this analysis was in fact redundant. It tried to show that VE PT more often exceeded a significance threshold than VE ENV, which is entirely unsurprising once it has been accepted that the median VE PT is substantially and statistically significantly larger than VE ENV. The revised manuscript now makes the point that VE PT is generally much larger than VE ENV more robustly with an added GEE analysis (see above), and this portion of the manuscript which tried to make the same point by looking at proportions of significantly PT ITD tuned vs ENV ITD tuned multi-units has been removed as it does not add extra value to the manuscript.

-Lines 486 - 496: Please consider discussing Müller et al. 2023 (doi.org/10.3389/fnins.2022.1021541) in this context.

We would like to thank the reviewer for drawing our attention to the relevance of this interesting paper to our Discussion. We now discuss our results in the context of the observations made by Müller et al. (2023). Please see the first paragraph in the discussion section "Comparison to previous electrophysiological work in CI animals" (lines 531-539).

Referee #2:

This report presents the remarkable observation that multi-unit responses in the rat's inferior colliculus are sensitive to 80-microsecond interaural differences (ITDs) in cochlear-implant pulse trains. The focus of the present study is on whether the sensitivity is dominated by ITD in the pulse timing or in the envelopes of the modulated pulse trains. The clear answer is that pulse timing dominates the ITD sensitivity, which accords with results of a behavioral study by this research team. One is concerned, however, that the present finding of microsecond-scale ITD sensitivity in the IC might have been corrupted by contamination of responses by electrical artifact. The artifact contamination would have had a greater effect on pulse-timing than on envelope ITD, which could explain the observed dominance of PT over ENV on ITD sensitivity. One needs to be convinced that the main result of the study is not due to contamination by electrical artifact before this report could be published.

General Comment: The stimulation and recording design of this study make it challenging to isolate and reject the electrical artifact. The biphasic electrical pulses had durations of ~123 microsec in the 0-ITD condition and 205 microsec in the +/- 80-microsec-ITD conditions. That means that the artifact occupies up to ~18% of the inter-pulse periods of 1111 microsec for the 900-pps rate and ~92% of the 222 microsec period for the 4500-pps rate. The output rate of the 41-microsec pulses would have been exactly at the 24.4-kHz input sample rate, meaning that details of

pulse shapes might have been subject to aliasing. The duration of the multi-phase artifact would have been 2 pulses longer for the 80-microsec-ITD condition than for the 0-ITD condition. That means that if any electric artifact were to get past the template subtraction and contribute to mean amplitude, the mean amplitude would be greater in the 80-microsec-ITD condition. Figure 3 gives the only examples of multi-unit amplitudes. In each of the cases in the lower row (showing PT ITD effects), amplitudes were greater for +/- 80 microsec-ITD conditions than for the 0-ITD condition. That non-monotonic ITD sensitivity fits well with an explanation based on longer artifacts for non-0-ITD conditions. In contrast, the non-monotonicity conflicts with one's expectation of ITD coding in the IC and with this research group's reports of behavioral discrimination of +80 versus -80 microsec; in 2 of the 3 PT plots, the peak AMUA values did a better job of discriminating + and - 80 microsec from 0 ITD than for discriminating +80 from - 80, which is what was required in the behavior protocol. The artifact explanation is not relevant to the upper row of the figure, in which the PT ITD was constant at 0 microsec.

We understand the reviewer's concern and appreciate that it is plausible to assume that different ITDs will lead to slightly different artifact shapes which might leak through the artifact filters to a differing extent. Indeed, the reviewer's comments have prompted us to perform additional tests on our artifact rejection and to refine our approach. Several changes were made to the manuscript which will hopefully convince the reader of the fact that contributions of residual artifacts to our results are negligible. Our original figure 3 (now Fig. 4) showed only three multi-units out of a sample of many hundreds. To give a larger (but still very sparse) sample of responses, we changed the figure to now contain six examples, and these now include one multi-unit (#5, bottom row, center) in which 0 PT ITD gave the largest, rather than the smallest, evoked response. Of course we are aware that presenting a counter example to the "mean amplitude would be greater in the 80 μ s ITD condition" observation that made this reviewer suspicious is not enough to generate confidence in the adequacy of our artifact rejection method, which is why further changes were made, as detailed below.

One might be more convinced of the effectiveness of the artifact rejection if the authors were to illustrate examples of templates and template subtraction for 0- and 80-microsec conditions and to show us how biological responses survive template subtraction. Such an illustration might also help explain the quality-control procedure given in lines 278-290, which this reader found difficult to understand.

After much reflection we concluded that showing examples of 0 μ s and 80 μ s templates and their subtractions, as suggested here, might be useful but would be unlikely to be enough. The artifact amplitudes are routinely many tens of times larger than the responses, the responses are very drawn out in time, and for the human eye it is very hard to see by looking at an example how many and which of the peaks one might observe in a waveform are residual artifact spikes rather than neural hash.

However, illustrating how the quality control procedure operates with a figure is an excellent suggestion. We have therefore added a new Figure 3 which shows examples of a traces for which artifact rejection was considered unsuccessful (A) or successful (B). We have to confess that the original description of our quality control procedure was poorly written and we thank the reviewer for flagging this. We have thoroughly revised this part. Together with the new Figure 3 we hope the quality control is now relatively easy to follow.

To address the more important point raised by the reviewer that we need to give our readers confidence that the artifact rejection operates in a manner that is “unbiased” in the sense that it does not lead to systematic, stimulus dependent differences in the amount of artifact residue in the recordings, we decided that a control experiment was be required, in which we tested our artifact rejection protocol on recordings from an agar block instead of a live animal. The physics and the signal processing for the agar block recordings are essentially identical to those from our live animal recordings, so they would suffer from the same stimulus-dependent leakage of artifacts into the recorded signal, and if the analysis pipeline is not adequate, then the recordings would show significant “neural tuning to stimulus ITD” even though the agar block lacks an auditory pathway. In the event, the VE by ITD in the “multi-unit recordings” taken from the agar block was negligible, no larger than expected from sampling variance given recording noise, and very small compared to the VE values seen in our physiological recordings. We now include these agar block recordings as Supplementary Figure 1 and describe them at the end of the “Artifact rejection” subsection of the Methods with the following text:

“In addition to this approach for identifying and excluding recordings for which the template based artifact rejection method appeared insufficient, we conducted a separate experiment to ensure that any residual artifact that might be contained in our recordings, would not introduce significant bias into our results. The concern here is as follows: the electrical artifact seen by the recording electrodes results from a summation of the effects created by the left and the right CI electrodes. As a consequence, the electrical artifact pulses created by stimuli with PT ITDs of +80 or -80 μ s, for which the left and right ear pulses occur in succession, will be broader in time than those with zero PT ITD, for which left and right ear pluses overlap precisely. It is conceivable that the filtering steps involved in our algorithm may not work equally well for each of the different stimulus artifact shapes created by -80, 0 or -80 μ s PT ITD stimuli respectively. In that case, even small amounts of artifact residue could add different amounts of background noise to the recordings for each PT ITD condition, and if they leak through to the response strength quantification described in the next section, they would introduce a bias into our estimates of ITD tuning strength. However, any such potential, stimulus parameter dependent “leakage” of artifacts into the recorded responses is solely a function of the physics of the stimulation and recording equipment and of the signal processing algorithms, and it is independent of any biological mechanisms. Consequently we can use “dummy”

recordings, in which the head of an experimental animal is replaced by a block of agar and physiological saline, to verify that signals recorded by our setup and processed by our analysis pipeline contain only negligible amounts of false-positive “neural response variance” that is attributable to variation in stimulus parameters. We conducted such a control experiment in which “responses” were recorded from a 1% agar in 0.9% saline block in which our 32-channel recording electrode and our binaural CI electrodes had been inserted in a configuration approximating that of our experimental animal preparation. The agar recordings were then processed and subjected to the same statistical analysis as the physiological data (see below). The results of these agar recordings can be found in Supplementary Figure 1. They confirmed that, following our artifact removal protocol, the amount of ITD tuning attributable to leaked, residual artifact was negligible.”

The new Supplementary Figure 1 is shown below for easy reference:

Supplementary Figure 1. Variance explained (VE) values for pulse timing ITD (PT ITD) and envelope ITD (ENV ITD) from a block of saline agar into which two CI electrodes and a 32-channel silicon electrode were inserted in order to mimic the in vivo measurements in CI rats. Recordings were performed just as for the experimental animals, including for all three envelope durations (from left to right: 10 ms, 50 ms, 200 ms) and at two pulse rates, 900 pps (upper row) and 4500 pps (lower row), and the agar control data were analysed and presented just as for Figure 5. Each dot shows the VE values computed for one of the 32 recording

channels. Values on the Y-axis indicate the percentage of VE explained by PT ITD. X-axis indicates the percentage of VE explained by ENV ITD. As expected, the VE values in this control experiment were close to zero, indicating that residual stimulation artifacts make at best only negligible contributions to the often sizeable VE PT values observed in the AMUA responses taken from our CI rats.

Specific comments:

Lines 76-77: "very limited ability" to provide PT ITD is an understatement. Present-day processors deliver no stimulus-relevant PT ITD.

Done, changed it now to "missing ability" (line 75).

Line 92: "the adjacent electrode" should be "adjacent electrodes"

Revised accordingly (line 91).

Line 96: "random numbers" is not exactly correct. The PT ITD delivered by such devices would be constant at some value unrelated to the sound-source ITD.

We would like to thank the reviewer for this valid objection and have tried to clarify and implement that comment and changed the sentence to "devices are not random in a strict sense but instead assume constant values that bear no relation to the sound-source ITD" (lines 95-97).

Line 150: "permanent access" here and later is an odd usage. "constant access" would be better.

Thank you for the comment. We changed that accordingly.

In the chronic CI cohort, how many hours of electrical stimulation and training/testing did each rat receive each day? How many days per week?

We added an elaboration on that now at the end of the section "2-AFC training and testing of the chronic CI cohort": "Animals were trained five days a week, with two training sessions per day and at least five hours between sessions. Each training session lasted between 20-30 minutes."

Artifact rejection: Please clarify whether or not a unique template was computed for each stimulus condition (including for each ITD).

It was a unique template for each condition. We completely rewrote the sub-section "Artifact rejection" in our Methods and the new description now also clarifies this point (see lines 288-290).

Lines 351-353: This sentence sounds as if one is defending against a claim that the examples were chosen with a bias against VE(ENV) conditions. It would be better just to say that the examples were selected to cover the ranges of VE values shown in later figures.

We changed that sentence and the whole paragraph style basically (lines 418ff) because we show more multi-units as requested (see previous response to major comments).

Please provide an illustration or statistic to show the distributions of multi units having minimum and maximum amplitudes at -80, 0, or +80 microsec PT ITD. Alternatively, one could show box plots of normalized mean AMUA at -80, 0, and +80 microsec. One finds it odd that all of the examples had max at +80 and min at 0 ITD.

The original examples in Fig. 4 (former Fig. 3) had been selected to show large, average and small $VE(PT)$ examples, and the fact that all three happened to have their highest AMUA peaks were coincidental. To better illustrate the diversity of AMUA responses, we now show more multi-units in Fig. 4. This now also includes examples in which maximal AMUA values at PT ITD other than +80 μ s. Note that our overall AMUA responses were calculated from the average, not the maximal AMUA amplitude.

What was the pulse rate for the examples in fig. 3?

The pulse rate was 900 pps and we added this information to the figure description of former Fig. 3, now Fig. 4.

In fig. 5, in all cases for the acute CI animals, $VE(PT)$ was higher for 4500 than for 900 pps (also shown in fig. 6). Greater ITD sensitivity at higher pulse rates is contrary to published results for normal hearing and contrary to likely physiological mechanisms. That result, however, is what one would expect if the $VE(PT)$ were driven by electrical artifact.

As previously mentioned, we tweaked the artifact rejection pipeline in the course of conducting an additional control experiment. Now there is no trend observable anymore showing " $VE(PT)$ was higher for 4500 than for 900 pps", looking at the

median values in Fig. 6. We hope the new Supplementary Figure 1 demonstrates that we can be confident that electrical artifacts make at best negligible contributions to $VE(PT)$.

Line 408: should be "sensitive IC units for each cohort". Here, you claim an observation (% of PT ITD units increases with duration) based on $N=3$, with 2 supporting the observation and one exception. One might soften that statement, like "% of PT ITD units generally increased".

The reviewer is correct and the statement would benefit from being softened down. However, upon revision, we determined that the analysis in question and therefore the whole paragraph was not essential and somewhat redundant like described previously. We have therefore removed the analysis and the paragraph in question from the manuscript.

Lines 475-476: Please remind us here that Schnupp et al. (2025) was a behavioral study.

We added this information as requested (line 513).

END OF COMMENTS

Dear Dr Schnupp,

Re: JP-RP-2026-290143R1 "Neurophysiological Sensitivity to Envelope and Pulse Timing Interaural Time Difference in Cochlear Implanted Rats with Different Hearing Experiences" by Shiyi FANG, Tim Fleiner, Fei Peng, Sarah Buchholz, Muhammad Zeeshan, Nicole Rosskothén-Kuhl, and Jan Wilbert Schnupp

Thank you for submitting your revised Research Article to The Journal of Physiology. It has been assessed by the original Reviewing Editor and Referees and has been well received. Some final revisions have been requested.

REVISION CHECKLIST:

Please upload two versions of your manuscript text: one with all relevant changes highlighted and one clean version with no changes tracked. The manuscript file should include all tables and figure legends, but each figure/graph should be uploaded as separate, high-resolution files. The journal is now integrated with Wiley's Image Checking service. For further details, see: <https://www.wiley.com/en-us/network/publishing/research-publishing/trending-stories/upholding-image-integrity-wileys->

image-screening-service

We look forward to receiving your revised submission.

Yours sincerely,

Nathan Schoppa
Senior Editor
The Journal of Physiology

REQUIRED ITEMS FOR REVISION

- You must start the Methods section with a paragraph headed Ethical approval (https://jp.msubmit.net/cgi-bin/main.plex?form_type=display_requirements#methods).

Research must comply with The Journal's policies regarding animal experiments (<https://physoc.onlinelibrary.wiley.com/hub/animal-experiments>) and adherence to these policies must be stated in the manuscript.

Authors should confirm in their Methods section that their experiments were carried out according to the guidelines laid down by their institution's animal welfare committee, including an ethics approval reference number. The Methods section must contain a statement about access to food, water and housing, details of the anaesthetic regime: anaesthetic used, dose and route of administration, and method of killing the experimental animals.

- Your paper contains Supporting Information of a type that we no longer publish, including supplementary tables and figures. Any information essential to an understanding of the paper must be included as part of the main manuscript and figures. The only Supporting Information that we publish are video and audio, 3D structures, program codes and large data files. Your revised paper will be returned to you if it does not adhere to our Supporting Information Guidelines.

- We invite you to include a Translational Perspective paragraph in your manuscript. This should be included in the main body of the manuscript after the Acknowledgements. It should describe the wider translational implications of the work, in plain English, for a broad scientific audience. Please use the following guidelines to prepare a Translational Perspective of your paper: https://jp.msubmit.net/cgi-bin/main.plex?form_type=display_requirements#authortranspersp. The Translational Perspective should not exceed 250 words in total and should be presented as a single paragraph. Abbreviations and technical terms must be defined as briefly and simply as possible the first time they are used, unless they are generally/easily understood, e.g. ECG, HIV/AIDS, K⁺ channel. Use language that can be understood by scientists or clinicians with a general knowledge of the topic addressed. Ensure the paragraph includes the hypothesis tested in the paper and accurately reflects the findings of the paper and the implications for future research. Please state the word count of the Translational Perspective paragraph.

- Papers must comply with the Statistics Policy: https://jp.msubmit.net/cgi-bin/main.plex?form_type=display_requirements#statistics.

In summary:

- If $n \leq 30$, all data points must be plotted in the figure in a way that reveals their range and distribution. A bar graph with data points overlaid, a box and whisker plot or a violin plot (preferably with data points included) are acceptable formats.
- If $n > 30$, then the entire raw dataset must be made available either as supporting information, or hosted on a not-for-profit

repository, e.g. FigShare, with access details provided in the manuscript.

- 'n' clearly defined (e.g. x cells from y slices in z animals) in the Methods. Authors should be mindful of pseudoreplication.
 - All relevant 'n' values must be clearly stated in the main text, figures and tables.
 - The most appropriate summary statistic (e.g. mean or median and standard deviation) must be used. Standard Error of the Mean (SEM) alone is not permitted.
 - Exact p values must be stated. Authors must not use 'greater than' or 'less than'. Exact p values must be stated to three significant figures even when 'no statistical significance' is claimed.
- Thank you for submitting your revised Research Article to The Journal of Physiology. It has been assessed by the original Reviewing Editor and Referees and has been well received. Some final revisions have been requested.
- Please include an Abstract Figure legend. An appropriate figure legend, which should not exceed 150 words in length, should be included in the main manuscript file.
-

EDITOR COMMENTS

Reviewing Editor:

Thank you for the thorough revisions. Both reviewers agree that your manuscript is now (almost) publishable in the Journal of Physiology. Please add the details about postoperative care to the methods section and also tend to the remaining minor wording issues raised by reviewer 2.

Senior Editor:

Thank you for sending your revised manuscript to The Journal of Physiology. The referees of your original manuscript were generally satisfied with the changes/additions that you made in response to their prior concerns. Prior concerns of the editors were also well-addressed, including the points raised about statistics. We believe that the manuscript is suitable for publication, pending your addressing a few remaining minor concerns. Most of these are described in the referee comments and reviewing editor's report.

I will also add that the authors should double check to make sure that "Fig. 5" that is referenced at the end of the Methods section is not meant to be Fig. 6 in the revised paper. Also, per Journal policy, supplementary figures are not allowed. The data there should be incorporated into either the main figures or into the text.

The referees, as well as the editors, believe that the study will be quite influential for the understanding of the limitations of cochlear implants in their sensitivity to interaural time differences and for potential improvements in these implants. We thus invite the authors to write a single-paragraph Translational Perspective on their study, which will be placed at the end of their article. We believe that such an addition will increase the impact and readership of the work.

See the REQUIRED ITEMS FOR REVISIONS list above.

Guidelines for Translational Perspectives are also provided in the Information for Authors.

REFeree COMMENTS

Referee #1:

The authors addressed my major comments and made adequate changes to the manuscript. These include improvements to the introduction, which now more clearly defines the motivation of the paper, and a more detailed discussion of the possible mechanisms behind superior PT ITD coding compared to ENV ITD coding. Furthermore, the authors provide a convincing justification that a detailed analysis of a potential onset effect is beyond the scope of the present study. The expanded mechanistic discussion now offers a useful basis for future investigations of this issue. Moreover, the authors increased the number of AMUA responses shown in Figure 4, which demonstrates the diversity of response patterns in the AMUAs. Together, these revisions satisfactorily address my major concern.

All minor concerns addressed in my previous comments were also given full consideration by the authors.

A final point that has not yet been raised concerns the post-operative procedures. The study is missing a description of post-operative care after cochlear implantation in the neonatally deafened + chronic CI group. Please provide these details in the relevant method section (e.g., were the animals given analgesia and/or anti-inflammatory agents after cochlear implantation?)

Referee #2:

I am generally satisfied with the authors' responses to my previous comments. I have only a few remaining comments and corrections.

Line 64: Would it be correct to add something to the Laback citation like: "; ITD thresholds were substantially higher at higher pulse rates"?

Line 71: "evitable" is rarely used in American English - I had to look it up. I suggest replacing evitable with something like "can be avoided".

Line 84: Here is "random PT ITDs" that I complained about in the first submission. I suggest changing it here to "tend to use pulses fired at a fixed rate independently in each ear that are unrelated to the acoustic stimulus ITD".

Line 96: This paragraph largely repeats content of the previous paragraph. I suggest deleting "are not random in a strict sense". Nothing has been said about randomness -- there is no reason to apologize for it.

Line 124: I see no reason for "a well described midbrain structure" and suggest deleting it.

Line 208, "Two electrode sites": What was the spacing between the two sites?

Line 379, "approach thinks of": Approaches don't think. I suggest changing to "approach treats".

END OF COMMENTS

The Journal of Physiology Decision on Our Manuscript: "Neurophysiological Sensitivity to Envelope and Pulse Timing Interaural Time Differences in Cochlear Implanted Rats with Different Hearing Experiences", *JP-RP-2025-290143*.

Dear Prof. Schoppa,

Dear Reviewers,

We would like to thank the editor for allowing us to submit a second revised version of our manuscript entitled "Neurophysiological Sensitivity to Envelope and Pulse Timing Interaural Time Differences in Cochlear Implanted Rats with Different Hearing Experiences". In addition, we thank the reviewers and the editor for their careful evaluation of our revised manuscript and for the constructive and encouraging feedback. We greatly appreciate the time and effort invested in assessing our work, as well as the positive acknowledgment of the additional analyses and revisions we implemented.

We have carefully addressed the remaining minor comments and made the corresponding changes in the manuscript. We believe that these revisions have further strengthened the clarity and quality of the paper. We have submitted a detailed reply to the reviewers' comments in the "Point-by-point response" attached below. Please also find our revised manuscript (tracked changes mode), in which we have highlighted all amendments to our manuscript.

Thank you again for your thoughtful guidance throughout the review process. We sincerely appreciate the opportunity to revise our work and look forward to your final decision.

Yours sincerely,

Shiyi Fang, Tim Fleiner, Fei Peng, Sarah Buchholz, Muhammad Zeeshan, Nicole Rosskothén-Kuhl and Jan W. Schnupp

REQUIRED ITEMS FOR REVISION

- You must start the Methods section with a paragraph headed Ethical approval (https://jp.msubmit.net/cgi-bin/main.plex?form_type=display_requirements#methods).

Research must comply with The Journal's policies regarding animal experiments (<https://physoc.onlinelibrary.wiley.com/hub/animal-experiments>) and adherence to these policies must be stated in the manuscript.

Authors should confirm in their Methods section that their experiments were carried out according to the guidelines laid down by their institution's animal welfare committee, including an ethics approval reference number. The Methods section must contain a statement about access to food, water and housing, details of the anaesthetic regime: anaesthetic used, dose and route of administration, and method of killing the experimental animals.

Please find at the beginning of the Methods section a paragraph titled “Ethical approval”, which provides information on the ethics committees that approved our study. Furthermore, ethical approval codes were provided, along with information on the animals’ sources, access to water and food, and housing statements. We also added a “Confirmation of compliance” statement in this paragraph. We provide information about humane killing (according to the Journal’s guidelines), the anesthetic regime and monitoring, post-operative care and recovery, surgical procedures, and terminal procedures further below in the revised Methods section.

- Your paper contains Supporting Information of a type that we no longer publish, including supplementary tables and figures. Any information essential to an understanding of the paper must be included as part of the main manuscript and figures. The only Supporting Information that we publish are video and audio, 3D structures, program codes and large data files. Your revised paper will be returned to you if it does not adhere to our Supporting Information Guidelines.

We would like to thank the Editor for this information and we decided that our supplementary figure is essential for the understanding of the paper and adds crucial information about the quality of our artifact rejection. Therefore, we moved the figure into the methods section to the end of the paragraph “Artifact rejection” (Fig. 4) and adjusted the subsequent figure numbers and descriptions accordingly.

- We invite you to include a Translational Perspective paragraph in your manuscript. This should be included in the main body of the manuscript after the Acknowledgements. It should describe the wider translational implications of the work, in plain English, for a broad scientific audience. Please use the following guidelines to prepare a Translational Perspective of your paper: https://jp.msubmit.net/cgi-bin/main.plex?form_type=display_requirements#authortranspersp. The Translational Perspective should not exceed 250 words in total and should be presented as a single paragraph. Abbreviations and technical terms must be defined as briefly and simply as possible the first time they are used, unless they are generally/easily understood, e.g. ECG, HIV/AIDS, K+ channel. Use language that can be understood by scientists or clinicians with a general knowledge of the topic addressed. Ensure the paragraph includes the hypothesis tested in the paper and accurately reflects the findings of the paper and the implications for future research. Please state the word count of the Translational Perspective paragraph.

We would like to thank the Editor for the invitation to include a 'Translational perspective' paragraph and we would like to take the opportunity to describe the wider translational implications of our work. We added this paragraph at the end of the manuscript, below the acknowledgments paragraph.

- Papers must comply with the Statistics Policy: https://jp.msubmit.net/cgi-bin/main.plex?form_type=display_requirements#statistics.

In summary:

- If $n \leq 30$, all data points must be plotted in the figure in a way that reveals their range and distribution. A bar graph with data points overlaid, a box and whisker plot or a violin plot (preferably with data points included) are acceptable formats.

- If $n > 30$, then the entire raw dataset must be made available either as supporting information, or hosted on a not-for-profit repository, e.g. FigShare, with access details provided in the manuscript.

- 'n' clearly defined (e.g. x cells from y slices in z animals) in the Methods. Authors should be mindful of pseudoreplication.

- All relevant 'n' values must be clearly stated in the main text, figures and tables.

- The most appropriate summary statistic (e.g. mean or median and standard deviation) must be used. Standard Error of the Mean (SEM) alone is not permitted.

- Exact p values must be stated. Authors must not use 'greater than' or 'less than'. Exact p values must be stated to three significant figures even when 'no statistical significance' is claimed.

We would like to thank the editor for highlighting this point and providing a summary of the statistical policy. We plotted our data (if $n > 30$) in a box plot fashion (Fig. 7) and described the respective features of the plot. In Figure 6, we plot the data from individual animals in a scatter plot fashion, as we want to show the distribution and clustering of the data. We stated the 'n' in the figure as well as pointed to references where to find the 'n' whenever possible (e.g. Table 1). Additionally, we provided the number of penetrations for each cohort and never reported SEM alone. We provided exact p-values in Figure 7 and throughout the results section according to the journal's guidelines.

We provide access to the raw data using this link: <https://auditoryneuroscience.org/dataShare>, which can be found in the 'Data availability statement' at the end of our revised manuscript.

Thank you for submitting your revised Research Article to The Journal of Physiology. It has been assessed by the original Reviewing Editor and Referees and has been well received. Some final revisions have been requested.

- Please include an Abstract Figure legend. An appropriate figure legend, which should not exceed 150 words in length, should be included in the main manuscript file.

We added a figure legend for the graphical abstract at the beginning of the manuscript.

EDITOR COMMENTS

Reviewing Editor:

Thank you for the thorough revisions. Both reviewers agree that your manuscript is now (almost) publishable in the Journal of Physiology. Please add the details about postoperative care to the methods section and also tend to the remaining minor wording issues raised by reviewer 2.

Senior Editor:

Thank you for sending your revised manuscript to The Journal of Physiology. The referees of your original manuscript were generally satisfied with the changes/additions that you made in response to their prior concerns. Prior concerns of the editors were also well-addressed, including the points raised about statistics. We believe that the manuscript is suitable for publication, pending your addressing a few remaining minor concerns. Most of these are described in the referee comments and reviewing editor's report.

I will also add that the authors should double check to make sure that "Fig. 5" that is referenced at the end of the Methods section is not meant to be Fig. 6 in the revised paper. Also, per Journal policy, supplementary figures are not allowed. The data there should be incorporated into either the main figures or into the text.

We thank the editor for pointing out this error. We confirm that the reference at the end of the Methods section should indeed be Fig. 6 (now Fig. 7 after

moving the supplementary figure into the Methods section). This has been corrected.

The referees, as well as the editors, believe that the study will be quite influential for the understanding of the limitations of cochlear implants in their sensitivity to interaural time differences and for potential improvements in these implants. We thus invite the authors to write a single-paragraph Translational Perspective on their study, which will be placed at the end of their article. We believe that such an addition will increase the impact and readership of the work.

We thank the editors and referees for the positive assessment of our work and for the suggestion to include a Translational Perspective. We have added a single paragraph at the end of the revised manuscript, highlighting the translational relevance of our findings for the design of future stimulation strategies.

See the REQUIRED ITEMS FOR REVISIONS list above.

Guidelines for Translational Perspectives are also provided in the Information for Authors.

REFEREE COMMENTS

Referee #1:

The authors addressed my major comments and made adequate changes to the manuscript. These include improvements to the introduction, which now more clearly defines the motivation of the paper, and a more detailed discussion of the possible mechanisms behind superior PT ITD coding compared to ENV ITD coding. Furthermore, the authors provide a convincing justification that a detailed analysis of a potential onset effect is beyond the scope of the present study. The expanded mechanistic discussion now offers a useful basis for future investigations of this issue.

Moreover, the authors increased the number of AMUA responses shown in Figure 4, which demonstrates the diversity of response patterns in the AMUAs. Together, these revisions satisfactorily address my major concern.

All minor concerns addressed in my previous comments were also given full consideration by the authors.

A final point that has not yet been raised concerns the post-operative procedures. The study is missing a description of post-operative care after cochlear implantation in the neonatally deafened + chronic CI group. Please provide these details in the relevant method section (e.g., were the animals given analgesia and/or anti-inflammatory agents after cochlear implantation?)

We have added a description of the analgesic and anti-inflammatory agents the rats received before the surgery and during the subsequent recovery period. Find these descriptions at the beginning and at the end of the second paragraph in “Cochlear implantation and measurement of electrical ABR” in our Methods section.

Referee #2:

I am generally satisfied with the authors' responses to my previous comments. I have only a few remaining comments and corrections.

Line 64: Would it be correct to add something to the Laback citation like: "; ITD thresholds were substantially higher at higher pulse rates"?

We thank the reviewer for this suggestion and have expanded the description of the Laback et al. (2015) findings to include information about pulse rate effects on ITD thresholds. The revised text (lines 62 to 65) now reads: 'To be precise, Laback *et al.* (2015) found a median ITD threshold of around 144 μ s, testing biCI patients with specialized experimental processors at low pulse rates of around 100 pulses per second and ITD thresholds were substantially higher at higher pulse rates (pps; Laback *et al.*, 2015).'

Line 71: "evitable" is rarely used in American English - I had to look it up. I suggest replacing evitable with something like "can be avoided".

We have replaced it with "can be avoided" as suggested (line 73).

Line 84: Here is "random PT ITDs" that I complained about in the first submission. I suggest changing it here to "tend to use pulses fired at a fixed rate independently in each ear that are unrelated to the acoustic stimulus ITD".

We thank the reviewer for catching this oversight. We have now revised the sentence (lines 83 to 86) to: 'In fact, commonly used coding strategies such as continuous interleaved sampling (CIS) or CIS derived strategies tend to use pulses fired at a fixed rate independently in each ear, which results in PT ITDs that are unrelated to the acoustic stimulus ITD and may introduce significant confusion.'

Line 96: This paragraph largely repeats content of the previous paragraph. I suggest deleting "are not random in a strict sense". Nothing has been said about randomness -- there is no reason to apologize for it.

Revised as requested (line 96 to 98).

Line 124: I see no reason for "a well described midbrain structure" and suggest deleting it.

The phrase has been deleted (line 123).

Line 208, "Two electrode sites": What was the spacing between the two sites?

We thank the reviewer for this suggestion. We have added the inter-site spacing information to our revised manuscript (line 212).

Line 379, "approach thinks of": Approaches don't think. I suggest changing to "approach treats".

Revised as requested (line 380).

END OF COMMENTS

Dear Professor Schnupp,

Re: JP-RP-2026-290143R2 "Neurophysiological Sensitivity to Envelope and Pulse Timing Interaural Time Difference in Cochlear Implanted Rats with Different Hearing Experiences" by Shiyi FANG, Tim Fleiner, Fei Peng, Sarah Buchholz, Muhammad Zeeshan, Nicole Rosskothén-Kuhl, and Jan Wilbert Schnupp

We are pleased to tell you that your paper has been accepted for publication in The Journal of Physiology.

Yours sincerely,

Nathan Schoppa
Senior Editor
The Journal of Physiology

IMPORTANT POINTS TO NOTE FOLLOWING ACCEPTANCE OF YOUR PAPER:

- **IMPORTANT NOTICE ABOUT OPEN ACCESS:** To assist authors whose funding agencies mandate immediate public access to published research findings, The Journal of Physiology allows authors to pay an Open Access (OA) fee to have their papers made freely available immediately on publication.

The Corresponding Author will receive an email from Wiley with details on how to register or log in to Wiley Authors where you will be able to place an order.

- You can check if your funder or institution has a Wiley Open Access Account here:
<https://authors.wiley.com/author-resources/Journal-Authors/open-access/author-compliance-tool.html>

- You can help your research get the attention it deserves! Check out Wiley's free Promotion Guide for best-practice recommendations for promoting your work at: www.wileyauthors.com/eoo/guide. You can learn more about Wiley Editing Services which offers professional video, design, and writing services to create shareable video abstracts, infographics, conference posters, lay summaries, and research news stories for your research at: www.wileyauthors.com/eoo/promotion.

- If you would like to receive our 'Research Roundup', a monthly newsletter highlighting the cutting-edge research published in The Physiological Society's family of journals (The Journal of Physiology, Experimental Physiology, Physiological Reports, The Journal of Nutritional Physiology and The Journal of Precision Medicine: Health and Disease), please click this link, fill in your name and email address and select 'Research Roundup':
<https://www.physoc.org/journals-and-media/membernews>

EDITOR COMMENTS

Reviewing Editor:

Congratulations on a nice piece of work!

Senior Editor:

Thank you for submitting your latest revised manuscript to The Journal of Physiology. All of the remaining minor concerns have been addressed and the work is now acceptable for publication. We also appreciate the well-written Translational Perspective that you added at the end. Congratulations on a nice study!